# Spatiotemporal variation of growth-stage specific concurrent climate extremes and their impacts on rice yield in southern China

Ran Sun<sup>1,2,3,4</sup>, Tao Ye<sup>1,2,3,4</sup>, Yiqing Liu<sup>1,2,3,4</sup>, Weihang Liu<sup>1,2,3,4</sup>, Shuo Chen<sup>1,2,3,4</sup>

O <sup>4</sup>Faculty of Geographical Science, Beijing Normal University, Beijing 100875, China

Correspondence to: Tao Ye (vetao@bnu.edu.cn)

Abstract. Increasing evidence highlights the disruptive effects of compound climate extremes on global crop yields under climate change. Existing studies predominantly rely on the whole growing-season scale and relative thresholds, and limit the ability to capture crop physiological sensitivities and yield responses that vary critically across growth stages. Here, we analyzed the spatiotemporal variations, dominant drivers, and potential impacts on the yields of concurrent heat-drought and chilling-rain events for single- and late-rice in southern China from 1981 to 2018. Specifically, we carefully distinguished three sensitive growth stages of rice and stage-specific climate stress types and thresholds based on rice physiology. Temporally, single-rice experienced a significant increase in concurrent heat-drought events, while late-rice experienced a modest rise in chilling-rain events. Spatially, the hotspots of concurrent heat-drought events varied greatly across the three growth stages. These spatial patterns are driven primarily by differences in crop phenology across locations, rather than by the occurrence of extreme climate conditions. The concurrent chilling-rain events of late-rice were widespread within the planting regions, with a higher incidence in certain areas. Path analysis identified heat stress as the primary driver of heat-drought impacts (particularly in jointing-booting and heading-flowering stages), whereas chilling and rain stress exerted comparable effects for late-rice. Our assessment of compound event impacts and sensitivity on rice yield revealed significant growthstage differences, with comparable yield losses from both concurrent heat-drought and chilling-rain events. Single-rice showed the highest sensitivity to heat-drought events during the grain filling stage, whereas the late-rice exhibited greater sensitivity during the heading-flowering stage. The historical impact on yield diverged markedly across growth stages, with the largest having occurred in the grain filling stage, particularly for heat-drought events. Our study provided important information on compound agroclimatic extremes, in the context of southern China's rice production system, and the results provide important information for risk management and adaptation strategies under climate change.

<sup>&</sup>lt;sup>1</sup>State Key Laboratory of Earth Surface Processes and Disaster Risk Reduction (ESPDRR), Beijing Normal University, Beijing 100875, China

<sup>&</sup>lt;sup>2</sup>Key Laboratory of Environmental Change and Natural Disasters, Ministry of Education, Beijing Normal University, Beijing 100875, China

<sup>&</sup>lt;sup>3</sup>Academy of Disaster Reduction and Emergency Management, Ministry of Emergency Management and Ministry of Education, Beijing 100875, China

## 1 Introduction

40

Compound climate extreme events, driven by the interaction of multiple drivers and/or hazards, often have more severe ecological and socioeconomic consequences than single events (Urban et al., 2018; Zscheischler et al., 2020). There is increasing concern regarding the future impacts of compound climate extreme events considering their projected increasing frequency and intensity (IPCC, 2022). Among the multiple potential impacts, agricultural production has received specific attention. The regional threats posed by these extreme events could further lead to global food security issues and the need to develop food system resilience (Chenu et al., 2017; Lobell and Gourdji, 2012; Trnka et al., 2014).

Previous studies have identified increasing trends in compound agroclimatic extremes, mostly in maize and wheat. Globally, analyses using diverse metrics, including growing—season precipitation—temperature anomalies (He et al., 2022), growing—season standardized anomalies of soil moisture and killing—degree—days (Lesk and Anderson, 2021), and Standardized Temperature Index (STI) with multiple drought indicators (i.e., scPDSI, SPI, and SPEI) (Feng et al., 2021), have consistently revealed intensified hot—dry extremes across major crops since 1950, with ~2% annual expansion of maize/wheat areas exposed to such events. Regionally, similar upward trends are seen in China's rainfed maize and wheat systems during 1980-2015 when assessed by percentiles of daily mean temperature and precipitation (Lu et al., 2018). However, analyses combining temperature indices (heating/freezing degree days) and drought indicators (SPI) or standardized drought—heat indices have revealed limited temporal trends despite the widespread spatial coverage of compound events since 1990 (Li et al., 2022; Wang et al., 2018).

The literature has also investigated the impact of compound agroclimatic extremes on yield, mostly focusing on compound heat and drought events (Lesk et al., 2021). Compound hot and dry summer conditions in the U.S. reduced soybean yields by two standard deviations, a sensitivity about four times larger than for heat alone and three times larger than for drought alone (Hamed et al., 2021). Another county—level study also showed that combined heat and drought events sharply reduce rainfed maize and soybean yields in the U.S. (Luan et al., 2021). In addition to concurrent hot—dry events, consecutive—dry—and—wet (CDW) extremes have been linked to yield losses: one analysis found that that the risk of yield loss caused by CDW extremes can be twice as high as that from individual wet and dry extremes (Chen and Wang, 2023).

Despite the growing recognition of compound climate extremes as critical threats to food security, critical knowledge gaps remain. First, while concurrent heat–drought events in staple crops have been extensively documented (Rötter et al., 2018), concurrent chilling and rain events have received little attention compared to heat–drought combinations. Second, most studies defineextremesusing relative statistical thresholds (e.g., percentiles of indicators) rather than crop– and stage–specific physiological thresholds, which may overlook important crop's biophysical sensitivities of by growth stage and event type (Kern et al., 2018). For example, rice faces different chilling thresholds of ≤ 17 °C at the booting stage and ≤ 20 °C at the grain filling stages (Zhang et al., 2014). Third, analyses focusing on the whole growing season can mask critical

sub-seasonal dynamics. For example, stress during the flowering stage can disrupt pollen viability and fertilization, while stress during the grain-filling stage can affect sucrose transport, which are all critical for yield formation (Sehgal et al., 2018; Xiong et al., 2016). Nevertheless, such stage-specific effects are seldom investigated independently. Additionally, quantitative analyses of yield losses under compound extreme in rice are limited.

Rice, as a critical staple crop for a large portion of the global population, deserves particular attention (Yu et al., 2024). Rice production in China includes single-rice in northeast China and in the Yangtze River Basin, and late-rice in southern parts of the country. The climate of these rice cropping systems varies substantially, from sub-tropical to warm temperate, and consequently the crop is exposed to a range of agroclimatic extremes. For single-rice, summer (July to September) is the highest temperature period in southern China and is prone to seasonal drought (Tan et al., 2020). At this time, single-rice in its jointing to flowering and maturity stage is vulnerable to the combined effects of heat and drought. From September to October each year, late-rice in its heading-flowering and grain filling stages is critically vulnerable to low temperatures, strong winds, and persistent rainy weather (Guo et al., 2020). These climate extremes compounded together are commonly referred to as "chilling-dew wind" and "continuous rain" events (Xie et al., 2016; Zhang et al., 2021). Climate change has driven more frequent and intensive extreme events for rice cultivation (He et al., 2022; Yu et al., 2024). The 2022 summer compound hotdry events in the Yangtze River Basin once induced considerable worry about the rice-based autumn grain production in southern China (Fu et al., 2024). In 2010, severe cold and rain stress caused the late-rice yield losses exceeding 1500 kg·hm<sup>-2</sup> in Hunan Province, central China (Lü and Zhou, 2018). Therefore, focusing on the compound climate extremes related to rice production in China could help add new wisdom about compound agroclimatic extremes to those reported about other staple crops.

This study aims to examine the spatiotemporal variations of concurrent compound extremes for single—and late—rice in southern China during 1981–2018, identify their underlying drivers, and quantify their impacts on yield. We focus on concurrent heat—drought events for single—rice, and concurrent chilling—rain events for late—rice, during the critical growth stages for each crop. The analysis uses crop—specific growth stages and physiological thresholds (detailed in Methods) to better capture the biophysical sensitivities of rice. Specifically, the study addresses the following questions: (1) How did the concurrent heat—drought and chilling—rain events change temporally and spatially in southern China's rice systems during 1981–2018? (2) To what extent are changes in compound severity driven by changes in individual climate factors? (3) What are the impacts of these concurrent events on rice yield? (4) How do the answers to the above question differ among different growth stages?

#### 0 2 Materials and Methods

## 2.1 Study area



Our study area covers the major rice–growing areas in southern China (Fig. 1). Local rice–growing systems include typical late–rice in the southeast and single–season rice (hereafter "single–rice") in the Yangtze River basin and southwestern China. Late–rice generally grows from July to November and is subjected to extremely low temperatures and continuous rain from September to October. Single–rice generally grows from June to November. Its heading–flowering stages overlap with the hottest season and are prone to drought owing to the hilly terrain of southern China (Tan et al., 2020). To best present the complicated temporal structure of climate extremes, both single– and late–rice were considered in our analyses.

**Figure 1. Raster samples of single–rice and late–rice growing areas.** Yellow grids indicate areas where single–rice is grown and blue grids indicate areas where late–rice is grown.

#### 2.2 Data



A gridded daily dataset containing daily mean temperature and precipitation was obtained from the CN05.1 dataset prepared by the Institute of Atmospheric Physics, Chinese Academy of Sciences (Wu and Gao, 2013). The CN05.1 is a gridded daily dataset based on interpolation from over 2400 observation stations in China, with spatial resolution of 0.25° latitude and 0.25° longitude. It is regarded as one of the best gridded climate forcing data in mainland China and has been widely used and tested in previous studies (Li et al., 2022; Zhu and Yang, 2020). The 0.25° gridded daily 0–10 cm soil moisture data were obtained from the VIC–CN05.1 surface hydrology dataset (Miao and Wang, 2020). The dataset was simulated by the latest variable

infiltration capacity (VIC) model and driven by pure station—based atmospheric forcings and high–resolution soil parameters based on field surveys. The modeled 0–10 cm soil moisture anomalies were highly correlated with in situ measurements (438 stations) during 2003–2016, with a mean R = 0.80.





We used two rice phenology datasets: rice agrometeorological station observations dataset (1981–2018) (CMA, <a href="http://data.cma.cn">http://data.cma.cn</a>) and the ChinaCropPhen1km dataset (2000–2019) (Luo et al., 2020). Rice agrometeorological station observations dataset was obtained from the China Meteorological Administration (CMA, <a href="http://data.cma.cn">http://data.cma.cn</a>), comprising rice phenological dates recorded by agrometeorological stations across China from 1981 to 2018. This dataset is considered the best quality crop phenology observation station dataset in China and has gained widespread usage (Chen et al., 2021; Liu et al., 2023; Zhang et al., 2022a). Each station systematically records the rice cropping type (single-rice or late-rice) and the corresponding dates of key phenological stages throughout the growing season, in accordance with the "Specifications for agrometeorological observation-Rice" developed in 2018. Rigorous checks and validation during the data preparation process resulted in the production of extremely accurate data on rice phenology, with an accuracy rate exceeding 95%. Records that exceeded twice the standard deviation were rejected to ensure the data quality (Zhao et al., 2016). The ChinaCropPhen1km dataset provides gridded rice phenology data at a 1-km spatial resolution for the period 2000–2019 (Luo et al., 2020). This data was derived based on Global Land Surface Satellite (GLASS) leaf area index (LAI) products. This dataset is superior to the previous one due to its spatially gridded format, but does not offer information before 2000. Both datasets were later fused to derive annual phenological dates from all rice-growing grids.

The annual spatial distribution data of single and late rice were obtained from a high–resolution distribution dataset of single–rice (Shen et al., 2023) and late–rice (Pan et al., 2021). The dataset provided a 10–m gridded distribution of single rice for 21 provinces in China and that of late rice for nine provinces in Southern China. The two datasets used a method that combined optical and synthetic aperture radar images based on the time–weighted dynamic time warping method. For single–rice, the data achieved an average overall accuracy of 85.23% across 21 provincial regions, based on 108,195 samples, with a mean R<sup>2</sup> value of 0.83 when compared to county–level statistical planting areas over three years. For late–rice, the identification accuracy reached 90.46% based on 145,210 survey samples. We took the data for 2020 as the southern China rice–growing area mask.

Historical gridded rice yield data were obtained from the AsiaRiceYield4km dataset (Wu et al., 2023) covering 1995 to 2015. The AsiaRiceYield4km dataset was generated by integrating multisource predictors into machine learning models, using inverse probability weighting to select the optimal model. It achieved high accuracy for seasonal rice yield estimation, with R<sup>2</sup> value of 0.88 and 0.91 for single and late—rice, and significantly outperformed existing models. Thus far, the dataset provides the longest time series covering all rice cultivation areas in China.

Owing to the difference in the spatial resolution of the above datasets, we harmonized those data to one base grid for later analyses. We used 0.25°×0.25° grids of the CN05.1 dataset as the base. Rice–growing area masks for single rice and late rice were then applied to the base grid map to mask valid rice–growing grids. As each 0.25°×0.25° climate grid covered many 10–m rice pixels, we kept climate grids with rice pixels ≥5% of the area of each climate grid. The final base map contained 2262 0.25°×0.25° grids for single–rice and 1383 0.25°×0.25° grids for late–rice (Fig. 1). For each grid, rice phenological dates were interpolated from station–observed dates using the co–kriging method with a Gaussian function, and the gridded phenology information from the ChinaCropPhen1km dataset as a covariate. Our interpolation effectively captured spatial variability characteristics and compensated for the sparse coverage of station observations in many areas. We also adjusted the resolution of AsiaRiceYield4km to the base grid using bilinear interpolation.

## 2.3 Individual extreme types and severity metrics

#### 2.3.1 Individual extremes considered




Three growth stages that were most susceptible to extreme stress were considered in this study: jointing–booting stage (#1), heading–flowering stage (#2) and grain filling stage (#3). The jointing–booting stage refers to the period from the first day of jointing to the last day before heading. The heading–flowering stage refers to the period from heading to flowering and generally lasts for 10 days. The grain filling stage refers to the period from the 11<sup>th</sup> day after heading to maturity. The exact dates of the different stages were obtained from phenological records for each year and station.

We considered four types of climate extremes known to impact rice yields: heat (H), drought (D), chilling (C) and rain (R).

Thresholds for these extremes were initially based on national and provincial standards. Our preliminary analysis showed that strictly adhering to these official thresholds led to a small sample size for a valid statistical analysis. Consequently, after a thorough literature review, we relaxed the thresholds of duration but reserved those for temperature/moisture. Finally, we specified thresholds for each climate extreme by growth–stage (Table 1), which were applied to daily climate data to screen the historical occurrence of these events.

Table 1 The thresholds of each individual extreme event.

| Rice type   | Growth stage                                                          | Climate extremes | Indicator & threshold: daily mean temperature (T/°C), daily total precipitation (PRE/mm), relative soil moisture (SM/%) |                      |  |
|-------------|-----------------------------------------------------------------------|------------------|-------------------------------------------------------------------------------------------------------------------------|----------------------|--|
| Single-rice | Jointing-booting (#1)<br>Heading-flowering (#2)<br>Grain filling (#3) | Heat             | T ≥ 33 °C                                                                                                               | ≥ 1 successive day   |  |
|             |                                                                       | Drought          | SM ≤ 75 %                                                                                                               | ≥ 10 successive days |  |
| Late-rice   | Heading-flowering (#2)                                                | Chilling         | T ≤ 20 °C                                                                                                               | ≥ 1 successive day   |  |
|             |                                                                       | Rain             | P ≥ 25 mm                                                                                                               | ≥ 1 successive day   |  |
|             | Grain filling (#3)                                                    | Chilling         | T ≤ 17 °C                                                                                                               | ≥ 1 successive day   |  |
|             |                                                                       | Rain             | P ≥ 25 mm                                                                                                               | ≥ 1 successive day   |  |

**Note:** The above thresholds are referenced from <NY/T 2915–2016>, Identification and classification of heat injury of rice; <NY/T 3043–2016>, Code of practice for field investigations and classification of rice seasonal drought stressess in southern—China; <NY/T 2285–2012>, Technical specification of field investigations and the grading of chilling damage to rice and; <DB5101/T 125–2021>, Indica rice weather stress level—continuous rain. NY/T is the *Agricultural Information Resource Classification and Coding Specification* in China. DB5101/T is the *Local Standard of Chengdu, Sichuan Province*. Thresholds for duration were relaxed from original standards to ensure adequate samples for later analyses.

## 2.3.2 Severity metrics for individual events



Here, severity (Haqiqi et al., 2021) was used to measure the stress imposed by individual extreme event. It was defined as the cumulative deviation from the threshold value of each stressor. Following this concept, heat stress (H) severity  $S_{H,g,t}$  at a given growth stage (g) in a given year (t) that meets the condition can be computed by the cumulative deviation of mean daily temperature (T) above its threshold ( $T_{base}$ ) for all the days (t) within this stage. We used 33°C as the base temperature (Table 1) in Eq. (1).

$$S_{H,a,t} = \sum_{i=1}^{n} |T_i - T_{base}| \quad (T_i \ge T_{base}) \tag{1}$$

Similarly, chilling stress severity  $S_{C,g,t}$  can be computed by the cumulative deviation of daily mean temperature (T) below its threshold ( $T_{base}$ ), for which we used 20 °C for heading–flowering stage and 17 °C for grain filling stage for one or more consecutive days in Eq. (2).

$$S_{C,a,t} = S_T = \sum_{i=1}^{n} |T_i - T_{base}| \ (T_i \le T_{base})$$
 (2)

Drought stress severity  $S_{D,g,t}$  can be computed by the cumulative deviation of soil moisture  $(SM_i) \le 75 \% (SM_{base})$  for 10 or more consecutive days in Eq. (3). Specifically, drought severity was calculated cumulatively from the first day that moisture fell below this threshold and only events lasting at least 10 consecutive days were retained for further analysis. The threshold of 10 days was applied based on physiological and agronomic relevance and experimental evidence (Amin et al., 2022; Barnaby et al., 2019). While extremely severe but brief droughts can be fatal, recent studies have also suggested that short–term drought triggers compensatory recovery post–stress, potentially accelerating grain filling without yield loss (Jiang et al., 2019; Li et al., 2005).

$$S_{D,q,t} = S_{SM} = \sum_{i=1}^{n} |SM_i - SM_{base}| \quad (SM_i \le SM_{base})$$
 (3)

Rain stress severity  $S_{R,g,t}$  can be computed by the cumulative deviation of daily total precipitation (PRE)  $\geq 25$  mm ( $PRE_{base}$ ) for one or more consecutive days in Eq. (4).

$$S_{R.a.t} = S_{PRE} = \sum_{i=1}^{n} |PRE_i - PRE_{base}| \quad (PRE_i \ge PRE_{base})$$

$$\tag{4}$$

For each grid, severity of heat, drought, chilling, and rain stress were computed by growth stage by using above equations.

## 190 2.4 Compound climate extremes types and severity metrics

#### 2.4.1 Compound climate extremes types



For compound climate extremes, we focus on cases where two types of stress occurred during the same growth stage, for example, simultaneous exposure to heat and drought during the jointing-booting stage of single-rice (Table 2). This definition aligns with the topological framework proposed by Zscheischler (Zscheischler et al., 2020) and is hereafter referred to as concurrent climate extremes. Specifically, for single-rice (Table 2), we defined three concurrent climate extremes: concurrent heat-drought events during the jointing-booting stage (H1D1), heading-flowering stage (H2D2), and grain filling stage (H3D3). A similar naming convention was applied to late-rice, which includes two concurrent climate extremes: concurrent chilling-rain events during the heading-flowering stage (C2R2) and grain filling stage (C3R3).

Table 2 The types of compound climate extremes.

| Single-rice | Climate extremes        | #1 Jointing-booting | #2 Heading-flowering | #3 Grain filling |
|-------------|-------------------------|---------------------|----------------------|------------------|
|             | Heat (H) & Drought (D)  | H1D1                | H2D2                 | H3D3             |
| Late-rice   | Climate extremes        |                     | #2 Heading-flowering | #3 Grain filling |
|             | Chilling (C) & Rain (R) |                     | C2R2                 | C3R3             |

Note: H: heat; D: drought; C: chilling; R: rain. #1: jointing-booting stage; #2: heading-flowering stage; #3: grain filling stage.

## 2.4.2 Compound severity metrics

To quantify the severity of concurrent climate extremes, we developed a copula–based framework for compound severity assessment (Li et al., 2021; Tavakol et al., 2020). This framework integrates (1) the modeling of marginal distributions and joint dependence using copula functions, by following Tootoonchi (Tootoonchi et al., 2022), (2) a correction procedure to account for years without any events, and (3) a transformation of the joint exceedance probability into a standardized severity index. The resulting metric enables consistent and comparable assessment of compound event severity.

## (1) Marginal and joint modeling using copulas

Let X and Y denote the univariate indices (severity) of climate extremes for the given growth stage in Table 2. The marginal distributions of the random variables X and Y are defined as u = F(X) and v = G(Y), respectively. To model the dependence structure between the two variables, we used copula theory to construct a bivariate joint distribution. The copula function C(u, v) captures the joint cumulative probability  $P(X \le x, Y \le y)$  and is expressed as:

$$P(X \le x, Y \le y) = C[F(X), G(Y)] = C(u, v) \tag{5}$$

A range of copula families were tested, and the best–fitting model was selected using goodness–of–fit tests (at a 0.05 significance level) and Bayesian Information Criterion (BIC) (Ribeiro et al., 2020; Salvadori et al., 2016). Models that cannot be rejected, based on p–values at the 0.05 significance threshold, are considered for final selection (Li et al., 2022; Sadegh et al., 2018). In our case, the Clayton copula was selected to construct the concurrent climate extremes.

## (2) Incorporating zero–severity samples into joint probability calculation

According to our definition, for years when there were no extreme events, severity scores (the calculated  $S_H$ ,  $S_D$ ,  $S_C$  or  $S_R$  values) will be "0". In the fitting process, samples with 0 values (u = 0 or v = 0) were not included, and should be taken back into account when we derive the joint exceedance probability. As our main quantity of interest is the joint exceedance probability P(X > x, Y > y), we apply the law of total probability to reconstruct the full joint exceedance probability by using  $P(A) = P(A|B) \times P(B)$ :

$$P_{S_{H_1}S_{D_1}} = P(S_{H_1} \ge x, S_{D_1} \ge y | x > 0, y > 0) \cdot P(x > 0, y > 0)$$
(6)

For instance, the joint distribution of concurrent heat–drought event across stages #1 can be fitted by using the severity of heat stress  $S_{H1}$  for stage #1 of all grids and all years together with that of the drought stress  $S_{D1}$  of stage #1.

Here, the conditional probability  $P(S_{H1} \ge x, S_{D1} \ge y | x > 0, y > 0)$  is computed from the copula as:  $1 - u - v + C_{H1D1}(u, v)$ , and the proportion of valid (non-zero) severity pairs is calculated as:  $\frac{n(x>0,y>0)}{N}$ , where n denotes the number of years when both severities are non-zero, and N is the total number of years. Therefore, the corrected joint exceedance probability becomes:

$$P_{S_{H1}S_{D1}} = P(S_{H1} \ge x, S_{D1} \ge y | x > 0, y > 0) \cdot P(x > 0, y > 0) = [1 - u - v + C_{H1D1}(u, v)] \cdot \frac{n(x > 0, y > 0)}{N}$$
 (7)

This adjustment ensures that the joint probability calculation reflects all years in the dataset, not just those included in the copula fitting.

(3) Inverse-transformation of Joint Exceedance Probability to Compound Severity Scores

To make the severity scores comparable across locations and compound types, we transformed the joint exceedance probability into a standardized z-score. This was done by applying the inverse standard normal distribution function  $\varphi^{-1}$ :

$$CS_{H1D1} = \varphi^{-1} [P_{S_{H1}S_{D1}}] \tag{8}$$

Higher CS values correspond to more severe compound events.




## 2.5 Contribution of temporal changes of Individual stress to compound events based on path analysis

We attempted to understand how the temporal changes in individual stress were attributed to compound climate extremes.

Specifically, we attempted to determine how the changes in compound severity (CS) of a specific concurrent climate extremes are related to the corresponding heat/chilling stress severity and drought/rain stress severity changes over time. Because there can be strong interactions between temperature and moisture, path analysis was conducted. A path analysis decomposes the interaction between the dependent and independent variables (correlation coefficients) into direct (direct path coefficients) and indirect (indirect path coefficients) based on a multiple linear regression, without requiring the variables to be independent of each other (Zhang et al., 2022b). It has been widely applied to estimate the magnitude and significance of hypothesized causal connections between dependent and independent variables when the effects of the variables are confounded (Zhang et al., 2022b, c; Yan et al., 2022).

We separated the system of correlations between the dependent variable and two corresponding independent variables to obtain the path coefficients. Taking single–rice as an example, the path coefficient of heat stress severity ( $S_H$ ) to compound severity ( $S_H$ ) to compound severity ( $S_H$ ), which was also the Pearson correlation coefficient between  $S_H$  and  $S_H$ , could be decomposed into direct and indirect effects by:

$$R_{S_H,CS} = P_{S_H,CS} + r_{S_H,S_D} P_{S_D,CS} \tag{9}$$

where,  $P_{S_H,CS}$  is the direct path coefficient of  $S_H$  on CS, and  $r_{S_H,S_D}$  is the Pearson correlation coefficient between the two independent variables,  $S_H$  and  $S_D$ . Thus,  $r_{S_H,S_D}P_{S_D,CS}$  is the indirect path coefficient of drought stress severity on CS.  $P_{S_H,CS}$  and  $P_{S_D,CS}$  are two standardized linear regression coefficients obtained by regressing CS on  $S_H$  and  $S_D$ . An F-test is conducted to test the statistical significance of the results, and the results of the path analysis were statistically significant when the p-value was < 0.05.

Based on the direct and indirect path coefficients, we calculated the determination coefficient (DC) to assess the explanatory power of individual and interactive climate stresses on compound events. For each climate variable (i.e., heat stress  $S_H$ , drought stress  $S_D$ , chilling stress  $S_C$ , and rain stress  $S_R$ ), the individual coefficient of determination was computed as  $DC_i = P_i^2$ , where

 $P_i$  is the total (direct plus indirect) path coefficient,  $i = S_H, S_D, S_C$  or  $S_R$ . To quantify the contribution from the cooperative interaction between two stresses, the co-determination coefficient was calculated as  $DC_{co} = 2P_i r_{ij} P_j$ , where  $r_{ij}$  is the correlation between variables i and j;  $i, j = S_H, S_D, S_C$  or  $S_R$ .  $DC_{co}$  can indicate the extent to which the interaction of two independent variables affected the compound extremes. The total explanatory power of all stresses, represented by the total coefficient of determination ( $DC_{total}$ ), was obtained by summing all individual and co-determination terms:  $DC_{total} = \sum DC_i + \sum DC_{co}$  Since  $DC_{total}$  captures both independent and interactive effects, its value may exceed 1, which reflects the cumulative explanatory power.

## 2.6 Assessment of compound climate extremes impact on yield

To evaluate the impact of concurrent climate extremes on rice yield, we used yield anomalies detrended from the historical yield time-series to isolate interannual variability from structural trends such as technological progress. The detrend method followed Wang and Zhang (Holly Wang & Zhang, 2003) and Ye (Ye et al., 2015), which fit log-linear regression models to historical yield time-series at each grid cell: —The yield at time t denoted by Y<sub>t</sub>, was modeled as:

$$\log\left(Y_{t}\right) = \beta_{0} + \beta_{1}t + \epsilon_{t} \tag{10}$$

Where  $\beta_0$  is the intercept and  $\beta_1$  represents the linear trend in the log-transformed yield.

The detrended yield anomaly  $Y_{d,t}$  was calculated as the residual from the regression:

$$Y_{d,t} = Y_t - \widehat{Y}_t \tag{11}$$

Where  $\widehat{Y}_t$  is the fitted yield at year t from the regression model.

To enable cross-grid and cross-year comparisons, we used standardized yield anomalies:

$$YA_t = \frac{Y_{d,t} - \mu}{\sigma} \tag{12}$$

Where  $YA_t$  is the standardized yield anomaly.  $\mu = \frac{1}{n} \sum_{i=1}^{n} Y_{d,t}$  is the mean of the detrended yield,  $\sigma = \sqrt{\frac{1}{n-1} \sum_{i=1}^{n} (Y_{d,t} - \mu)^2}$  and n-1 is used instead of n to provide an unbiased estimate of the population standard deviation.

To formally characterize the relationship between standardized yield anomalies and compound climatic stress, we employed a simple linear regression model. For each growth stage, the standardized yield anomaly (YA) was regressed on the corresponding compound severity (CS) value:

 $285 \quad YA_t = \gamma_0 + \gamma_1 \cdot CS + \delta \tag{13}$ 

where  $YA_t$  is the standardized yield anomaly (detrended and normalized, see section 2.6), CS is the compound severity,  $\gamma_0$  is the intercept representing the expected yield anomaly when compound stress is absent,  $\gamma_1$  represents the yield loss per unit increase in compound severity and  $\delta$  is the error term. The regression model is fitted exclusively using observations where  $YA_t 

Figure 2. Annual compound severity of concurrent compound events during 1981–2018. Panels (a–c) show the concurrent heat–drought events in single–rice during jointing–booting#1 (H1D1), heading–flowering#2 (H2D2), grain filling stages#3 (H3D3). Panels (d–e) show the concurrent chilling–rain events in late–rice during heading–flowering#2 (C2R2), grain filling stages#3 (C3R3). \* and \*\* indicate statistically significant at the significance levels of 0.05 and 0.01, respectively.

## 3.2 Spatial distribution of compound climate extremes



To characterize the spatial distribution of severity, the average severity was calculated across all years in which occurrences were recorded. Specifically, the annual compound severity for each type of concurrent climate extremes was averaged within each grid cell to identify and map spatial hotspots (Fig. 3). The patterns were clear and contrasting. The average compound

severity for concurrent heat-drought events covered a limited growing area, whereas that for chilling-rain events was widespread.

Hotspots of high—compound severity grids for concurrent heat—drought events differed largely among the three types (Fig. 3a—c). H1D1 (heat—drought events during jointing—booting stage) were concentrated in coastal areas, H3D3 (grain filling—stage events) were mainly concentrated in inland China and H2D2 (flowering—stage events) were mainly distributed between these two regions. Specifically, H1D1 were mostly concentrated in the lower reaches of the Yangtze River (East China region), while H3D3 were concentrated in the eastern part of the Sichuan—Chongqing area. H2D2 showed a clustered occurrence in central Anhui, eastern Hunan, and eastern Sichuan.


Figure 3. Spatial distribution of compound severity for concurrent climate extremes during 1981–2018. Panels (a–c) show concurrent heat–drought events in single–rice during jointing–booting#1 (H1D1), heading–flowering#2 (H2D2), and grain filling stages#3 (H3D3). Panels (d–e) show concurrent chilling–rain events in late–rice during heading–flowering#2 (C2R2), and grain filling stages#3 (C3R3). Shading represents compound severity (unitless index), with darker colors indicating higher stress severity.

Unlike heat—drought events, concurrent chilling—rain events were widespread within the planting regions, with a higher incidence in certain areas (Fig. 3d and 3e). Hotspots of C2R2 (chilling—rain events during heading—flowering stage) were mostly concentrated in the southern parts of the study area, hilly regions to the south of Hunan and Jiangxi, and eastern Guangxi. The hotspots moved northward in C3R3 (chilling—rain events during grain filling stage), reaching the northeastern part of the

study area, occurring in Hubei, Anhui, Zhejiang, and hilly regions in southern Hunan province where the altitude is relatively high.

#### 3.3 Effects of individual stress severity on concurrent climate extremes






We took the path coefficient as the relative sensitivity of CS (compound severity) to  $S_H$  and  $S_D$  for single–rice,  $S_C$  and  $S_R$  for late-rice. For three types of the concurrent heat-drought events, the direct path coefficient for heat stress severity  $(P_{SH,CS})$  and drought stress severity  $(P_{SD,CS})$  were both positive (Fig. 4a), indicating that the changes in the severities of heat and drought stress both contributed to increasing the compound severity. The contribution of  $S_H$  was much larger than  $S_D$  in stage#1, but slightly smaller in stage#3. Considering that the distribution of spatial hotspots for concurrent heat-drought events varied markedly across three growth stages (Fig. 3a-3c), the pattern also suggests the regional difference of relative contribution. In the lower-reaches of the Yangtze River Basin (where H1D1 and H2D2 occurred), heat stress was a greater determinant of concurrent heat-drought events than the drought stress, while in the eastern Sichuan Basin (where H3D3 occurred), the influence of drought stress exceeded slightly the influence of heat stress.

For single-rice, the total determination coefficient,  $DC_{total}$ , which indicates the total effect of the two independent variables on the dependent variable, was similar across concurrent heat-drought events (median around 0.9) (Fig. 4c). The single-factor determination coefficients ( $DC_{SH,CS}$  and  $DC_{SD,CS}$ ) indicated that the severity of heat stress affected the change of concurrent climate extremes to a greater extent than the severity of drought stress in H1D1 and H2D2, with a similar pattern observed for the path coefficients ( $P_{SH,CS}$ ,  $P_{SD,CS}$ ). The median  $DC_{co}$  was around 0.3, which indicated that the two variables are not independent and positively correlated. It is worth noting that the median of  $DC_{co}$  is higher than the median of  $DC_{Sp,CS}$  in H1D1 and H2D2, which may result from the dominant effect from heat stress on concurrent heat-drought events in jointing-booting stage (H1D1) and heading-flowering stage (H2D2).

The pattern of the effects of chilling and rain stress severity on concurrent chilling-rain events for late-rice was very different 350 to that of heat-drought events (Fig. 4b). Both chilling and rain stress severity had a strong direct effect on the changes in climate extremes, with chilling having a slightly larger effect in C2R2 and rain having a slightly larger effect on C3R3. This pattern was also supported by the DCs of individual variables ( $DC_{S_{C},CS}$  and  $DC_{S_{R},CS}$ ) (Fig. 4d).  $DC_{co}$  was almost 0 for both growth stages (Fig. 4d), due to the very small indirect coefficient, indicating that there was little correlation between the two stresses in concurrent chilling-rain events. That means the interactive effects of temperature and moisture had quite small influence on the changes observed in concurrent chilling—rain events for late—rice.

Figure 4. Boxplot of the path analysis of climate factors on the duration of concurrent climate extremes during 1981–2018. Only relationships that passed the *F*-test at the 0.01 significance level are presented. Panels (a, c) show the path coefficient and determination coefficient of concurrent heat-drought events in single-rice during jointing-booting#1 (H1D1), heading-flowering#2 (H2D2), grain filling stages#3 (H3D3). Panels (b, d) show the path coefficient and determination coefficient of concurrent chilling-rain events in late-rice during heading-flowering#2 (C2R2), grain filling stages#3 (C3R3).

## 3.4 Impact on yield of compound events



We used the linear regression model described in Section 2.6 to examine the relationship between compound severity and standardized yield anomaly across different growth stages, resulting in five statistical models for various compound events and stages. These models provide quantitative measures of the stage-specific sensitivity of rice yield to compound climatic stress. Figure 5 presents the fitted data points and the regression trend lines to visually illustrate the models. For each regression, we reported the slope ( $\beta_0$ ), intercept ( $\beta_1$ ), and significance level. To emphasize the magnitude of yield loss (negative yield

anomalies) under severe compound stress (negative values), the axes in Figure 5a–e were restricted to negative ranges. Five types of concurrent extreme events were examined: H1D1, H2D2, H3D3 (heat–drought), and C2R2, C3R3 (chilling–rain).

For heat–drought events on single–rice, the highest average yield loss occurred during grain filling stage (H3D3) (Fig. 5f). This phenomenon was determined by the combined effects of historical event severity, frequency, and spatial extent. Regression analysis (Fig. 5a–c) revealed significant positive relationships between compound severity and yield loss across all growth stages. Rice yield showed the largest sensitivity in the grain filling stage (H1D1,  $\beta_1 = 0.29$ , p 

Figure 5. Relationship between compound severity and standardized yield anomaly during 1995–2015. Panels (a–c) show concurrent heat–drought events for single–rice during jointing–booting#1 (H1D1), heading–flowering#2 (H2D2), grain filling stages#3 (H3D3). Panels (d–e) show concurrent chilling–rain events for late–rice during heading–flowering#2 (C2R2), grain filling stages#3 (C3R3). \*\*\* indicates statistically significant at the significance levels of 0.001.

#### 4 Discussion





## 4.1 Divergent spatial distribution patterns yet increasing temporal trends of concurrent events for rice

We revealed the spatiotemporal variation of concurrent compound extremes for single—and late—rice in southern China, using growth—stage—specific physiological thresholds for temperature and moisture (either soil moisture or precipitation). This approach minimizes uncertainties inherent in applying uniform thresholds across the entire growing season. For example, the spatial difference in the hotspots of concurrent heat—drought events of single—rice would not have been identified if we conducted evaluations over the entire growing—season. For the chilling stress to late—rice, the different effects of extremes at the heading—flowering and grain filling stages would not have been distinguishable if only one single temperature threshold was used to screen the whole growing—season. The consideration of a growth—stage—specific threshold enabled us to distinguish the different spatial and temporal characteristics of concurrent climate extremes in different stages for single—rice and late—rice.

Temporally, we found a statistically significant increasing trend in the compound severity of concurrent heat—drought events in southern China. The concurrent chilling—rain events for late—rice had a weak increasing trend, which was insignificant. The result was consistent with the increasing frequency of concurrent heat—drought events reported in previous studies. For example, increasing trends for concurrent heat—drought events in the main crop production areas since 1980 have also been reported by He (He et al., 2022), Zhang (Zhang et al., 2022b) and Lu (Lu et al., 2018). For chilling—rain events in late—rice, Liu (Liu et al., 2013) also reported that the frequency of chilling events in rice during the period 2001–2011 was higher than that in 1990–2000. They suggested that despite the increase in mean climatic temperatures, the occurrence of chilling events in rice did not decrease, but instead showed a gradually increasing trend. This pattern was also consistent with our findings.

Spatially, we found that concurrent heat–drought events occurred only in specific regions in each of the three growth stages of single–rice, and coincided with the occurrence of heat stress in each growth–stage (Fig. A2). These spatial differences could mainly be attributed to regional differences in rice phenology rather than regional high–temperature events. That said, high temperatures in July and August in southern China enacted the precondition for heat events, and the dates of the susceptible growth–stage eventually determined the final period of exposure to concurrent events. For example, the single–rice transplanting date was 30 days earlier (day of the year, DOY 174–198) in the upstream than in the lower Yangtze River basin (DOY 207–232). When the single–rice in Chongqing entered the grain filling stage, rice in the middle and lower reaches of

the Yangtze River just entered the jointing-booting stage. Consequently, concurrent heat-drought events had a higher frequency in the later growth-stage in the upstream than in the downstream.

Similarly, the late–rice heading date was 20 days earlier in the northern part of study area (DOY 255 in Hubei, Hunan, Anhui and Zhejiang) than in the southern part (DOY 273 in Guangdong, Guangxi and Hainan). In October, the late–rice in the northern part was mostly in the grain filling stage, whereas in the southern region, due to later planting dates, it was mostly in the heading–flowering stage. Consequently, southern late rice is more susceptible to the impact of chilly and rainy conditions caused by the southward movement of cold air from the north, which converges with warm and moist air currents in the south, leading to low–temperature and continuous rain days. This finding further emphasized the importance of using growth–stage–specific thresholds, which allowed the exact spatiotemporal overlap of climate extremes and susceptible growth stages to be captured.

## 4.2 The predominance of individual stress in driving concurrent events





Path analysis identified the relative contribution of individual stress to compound severity and found differences by growth stage. For instance, individual heat stress had a greater direct effect than drought stress on heat–drought events during jointing–booting (H1D1) and heading–flowering (H2D2) of single–rice, but this pattern was not apparent in heat–drought events during grain filling (H3D3). For concurrent chilling–rain events of late–rice, the effects of chilling and rain stress were comparable, with a slightly larger effect of chilling in C2R2 and a greater effect of rain stress in C3R3.

Previous studies on the factors driving changes in climate extremes have reported divergent results. Bevacqua (Bevacqua et al., 2022) speculated that precipitation trends determined the future occurrence of concurrent heat—drought events. This is because future local warming would be sufficiently frequent that future droughts would always coincide with moderate heat extremes, and consequently, the changes in drought frequency would become the modulating factor. In this study, concurrent heat and drought events in the joint—booting stage (H1D1) and in the flowering stage (H2D2) mainly occurred in the middle—lower Yangtze River Basin. The spatial distribution of single extreme events (Figure A1) showed that drought stress exhibited broad spatial coverage and higher severity across this region (Fig. A2 d, e). In contrast, heat stress was concentrated within limited areas (Fig. A2 a, b). Consequently, when heat stress occurred, it had a higher likelihood of coinciding with drought conditions, thereby forming concurrent heat—drought events. This spatial dichotomy highlights the fact that heat stress emerges as the dominant driver of concurrent heat—drought events, where its localized intensification, superimposed on drought conditions, triggers compound cascading effects. However, heat stress during grain filling stage in the Sichuan and Chongqing regions was slightly more severe than drought. (Fig. A2 c, f), thus, the heat in this region has a slightly higher impact.

The results of the path analysis also showed a correlation between the heat stress and drought stress of the concurrent heat–drought event (Fig. 4c,  $DC_{co}$ ). Previous studies have shown that enhanced dry–hot dependence can lead to more frequent concurrent heat–drought events (Hao and Singh, 2020; Zscheischler and Seneviratne, 2017). The combination of these processes leads to a strong negative temperature–soil moisture correlation, which can be explained by two pathways: land–atmosphere feedbacks and weather–scale correspondence between clouds and incoming shortwave radiation. Specifically, soil moisture deficits caused by low precipitation can lead to reduced evaporative cooling, along with increased sensible heat fluxes and higher surface air temperatures. High temperature anomalies accelerate evapotranspiration, which further depletes soil moisture (Liu et al., 2020; Miralles et al., 2019). In addition, low levels of cloudiness associated with low precipitation (and subsequent soil moisture deficits) tend to enhance incoming shortwave radiation, which leads to higher surface air temperatures (Berg et al., 2015). For chilling–rain events for late–rice, our results also indicated a weak individual chilling and rain correlation (Fig. 4d,  $DC_{co}$ ). However, compared with heat–drought events, the relationships behind chilling–rain events have largely been ignored in previous studies, and the underlying mechanism requires further investigation (Trotsiuk et al., 2020).

#### 4.3 The sensitivity of yield to concurrent events





Our study evaluated the historical impact on yield and its sensitivity of concurrent climate extremes across different sensitive growth stages and found comparable yield losses from concurrent heat–drought and chilling–rain events (Fig. 5a–e). Yield sensitivity also exhibited comparable values between heat–drought events (0.29 on average) and chilling–rain events (0.19–0.37). This comparable effect is due to the disruption of physiological processes, such as photosynthesis and nutrient uptake, while increasing pest and disease risks caused by chilling or excessive rainfall (Arshad et al., 2017; Fu et al., 2023; Jiang et al., 2010). Therefore, our results add important evidence about the impact of compound chilling–rain on rice yield, to those that have reported heat–drought events on crops such as maize and soybeans (Luan et al., 2021; Seneviratne et al., 2010).

Different impacts of heat–drought events on yields were also evident across growth stages, with the highest average yield loss observed during grain filling stage (H3D3) (Fig. 5f). Spatial distribution patterns of compound events indicate that H3D3 was concentrated in the Sichuan–Chongqing region (Fig. 3c). These losses likely resulted from the combined effects of regional exposure, climate interactions and local infrastructure limitations. Variations in regional climatic conditions can lead to differential yield responses to extreme events across geographical areas (Li and Tao, 2023). In the Sichuan–Chongqing hotspot, the concentration of heat–drought events was amplified by topography–driven vapor pressure deficit anomalies (Zhu et al., 2024), which intensified moisture stress and ultimately caused substantial yield declines. Moreover, the region's hilly terrain makes the development of irrigation infrastructure challenging (Ye et al., 2012), and rice cultivation here depends heavily on precipitation. Consequently, under persistent hot and dry conditions, the lack of irrigation facilities can further exacerbate yield losses (Hao et al., 2023).

Rice sensitivity to compound events also differed substantially according to the growth stage. Specifically, single–rice showed the highest sensitivity to heat–drought events during the grain filling stage, followed by the heading–flowering and jointing–booting stages. Late–rice exhibited greater sensitivity during the heading–flowering stage than during the grain filling stage. These growth–stage–specific patterns may be attributed to the physiological vulnerabilities of rice at different growth stages and the mechanisms by which climatic stressors exert their effects. Although experimental studies explicitly revealing the mechanisms of yield reduction under compound events remain limited, plausible explanations can be inferred from the physiological responses of rice to individual stressors. For instance, heat stress during the grain filling process inhibits grain starch biosynthesis and shortens the grain filling duration, leading to reduced grain weight and yield (Cao et al., 2008; Tenorio et al., 2013). Drought negatively impacts photosynthetic rate and chlorophyll content, while drought occurring during the grain filling stage reduces the 1000–grain weight, ultimately leading to yield loss (Amin et al., 2022). Chilling stress during the heading–flowering stage impairs rice yield by inhibiting spikelet opening, inducing spikelet sterility, and potentially leading to spikelet abortion and incomplete panicle exertion (Arshad et al., 2017; Suh et al., 2010). Rain stress exerts a physical disturbance on pollination, thereby reducing the number of filled grains per panicle. Additionally, the overcast conditions associated with rain stress severely impair photosynthetic assimilation in rice (Luo et al., 2018; Proctor, 2023).

#### 4.4 Limitations







Our study was limited by the length of the time—series of data. Agrometeorological station data were only available up to 2018, and recent years that had experienced the most pronounced warming (IPCC, 2021) were therefore not included in the analysis. In particular, the severe concurrent heat—drought event in southern China in 2022 had a substantial impact on rice production (Hao et al., 2023). The absence of above data might have led to underestimates of the temporal trend and yield impact. We focused on concurrent climate extremes only in this research. However, climate extremes can occur consecutively in different growth stages (Zscheischler et al., 2020). Several studies have discussed the impact on yield of switches of dry—and—wet in different stages of rice growth (Chen and Wang, 2023). Due to limited sample size, other types of compound climate extremes (like consecutive climate extremes, where rice is impacted by one event at one growth—stage, and by another at a different growth—stage) were not discussed in this study, but require future investigation, including its spatiotemporal variation, possible physical compound mechanisms, and the underlying processes of yield loss.

## **5 Conclusions**

In this study, we investigated the spatiotemporal variation of concurrent compound extremes for single— and late—rice in southern China and their underlying climate drivers, by distinguishing stage—specific climate stress types and thresholds based on rice biology. Temporally, our results indicated a significant increasing trend of concurrent heat—drought events for single—rice and a slight increasing trend for concurrent chilling—rain events for late—rice. Spatially, the hotspot distributions of concurrent heat—drought events varied greatly across the three growth stages, being concentrated in regions from the upper—

middle to the middle—lower reaches of the Yangtze River. These spatial patterns are driven primarily by differences in crop phenology across locations, such as the timing of flowering was earlier in the upstream than in the lower Yangtze River basin, rather than by the spatial distribution of extreme climate conditions. The concurrent chilling—rain events of late—rice were widespread within the planting regions, with a higher incidence at higher altitudes and latitudes. Path analysis suggested that heat stress had a larger direct effect than drought on compound severity, particularly in H1D1 and H2D2. For concurrent chilling—rain events of late—rice, the effects of chilling and rain stress were comparable. The assessment of compound event impacts and sensitivity on rice yield revealed significant growth—stage—specific differences, with comparable yield losses from both concurrent heat—drought and chilling—rain events.


Our results provided critical insights into the comprehensive impacts of compound events on rice production and established a scientific foundation for developing targeted adaptation strategies. A straightforward extension of the present study is to project the future occurrence and severity of compound extremes for rice, and their future impact on yield, for risk management and adaptation purposes. Such a projection requires quantitative vulnerability functions or growth model simulations of compound extreme events. To increase the capability of the models, controlled experiments and field observations are needed to improve our understanding of the impact of compound extremes on rice.

## Appendix A: Additional Figures


Figure A1. Copula cumulative distribution functions as 3D surface of *u* (heat or chilling severity) and *v* (drought or rain severity) for concurrent heat–drought events during jointing–booting#1 (a, H1D1); heading–flowering#2 (b, H2D2); grain filling stages#3 (c, H3D3) and concurrent chilling–rain events during heading–flowering#2 (d, C2R2); grain filling stages#3 (e, C3R3).

Figure A2. Spatial distribution of single heat and drought extreme events of rice for the period of 1981–2018. Each subgraph represents the heat stress severity (a–c) and drought stress severity (d–f).

#### 525 Author contributions

TY designed the research. Material preparation and data collection were performed by RS, YL, WL and SC. RS finished the data analysis and drafted the manuscript and TY revised it. All authors commented on previous versions of the manuscript. All authors read and approved the final manuscript.

## Data availability

The CN05.1 dataset (Wu and Gao, 2013) and VIC–CN05.1 surface hydrology dataset (Miao and Wang, 2020) were obtained by contacting <a href="mailto:wangjun@mail.iap.ac.cn">wangjun@mail.iap.ac.cn</a>. The ChinaCropPhen1km dataset (2000–2019) was publicly available at <a href="https://doi.org/10.6084/m9.figshare.8313530.v7">https://doi.org/10.6084/m9.figshare.8313530.v7</a> (Luo et al., 2020). The high–resolution distribution dataset of single-rice was publicly available at <a href="https://www.scidb.cn/detail?dataSetId=b07f90ea5f0c4e359fa4119a0030f9da">https://www.scidb.cn/detail?dataSetId=b07f90ea5f0c4e359fa4119a0030f9da</a> (Shen et al., 2023) and laterice was publicly available at <a href="https://nesdc.org.cn/sdo/detail?id=6195c2f07e2817528307c465&subjectcode=1315">https://nesdc.org.cn/sdo/detail?id=6195c2f07e2817528307c465&subjectcode=1315</a> (Pan et al., 2021). The AsiaRiceYield4km dataset (1995–2015) was publicly available at <a href="https://doi.org/10.5281/zenodo.6901968">https://doi.org/10.5281/zenodo.6901968</a> (Wu et al., 2023).

## Code availability

The code is available from the corresponding author upon reasonable request.

## **Competing interests**

The authors have no relevant financial or non-financial interests to disclose.

#### Acknowledgments

This study has been financially supported by National Natural Science Foundation of China (NSFC. 42171075), and the project jointly funded by National Natural Science Foundation of China (NSFC. 72261147759) and the Bill & Melinda Gates Foundation (2022YFAG1004).

AI tools were used for sentence and format checking.

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
