# Peer review of "Spatiotemporal variation of growth-stage specific concurrent climate extremes and their impacts on rice yield in southern China"

_EGUsphere, 2025_

## Author Comment (AC1)

**Title: Spatiotemporal variation of growth-stage specific concurrent climate extremes and their yield impacts for rice in southern China**

**Response to Reviewer Comments (RC1):**

**'Comment on egusphere-2025-1393', Anonymous Referee #1, 20 May 2025**

The manuscript presents a well-designed and timely study on the correlation between compound climate extremes and rice yields in southern China, with clear relevance to climate change adaptation. The authors leverage growth-stage-specific physiological thresholds, multi-source gridded data, and compound severity metrics to offer new insights into how concurrent heat-drought and chilling-rainy events affect rice production. This work makes an important contribution in the construction of metrics for compound stressors. However, several points require clarification, and improvements in structure and presentation would significantly improve the manuscript.

RE: Thank you so much for your comments and suggestions on our manuscript. We have responded to the comments and suggestions point-by-point below (in blue).

**Major Comments:**

**RC1.1** Ambiguity in Drought Stress Severity Definition. The calculation of drought severity appears to exclude events shorter than 10 days, regardless of intensity. Please clarify whether severity is accumulated continuously or only calculated if a 10-day event threshold is met. For rice, a very low soil moisture period, even for a week, can be fatal. Justification for this duration cutoff should be provided, ideally based on physiological or agronomic evidence.

RE: Thank you for the question. The calculation of drought severity accumulates from Day 1 (the onset of soil moisture falling below the defined threshold). However, we retain and analyze only events persisting for $\geq 10$ consecutive days. The threshold of ten days was applied based on physiological and agronomic relevance and experimental evidence.

Drought development in field environments (especially paddies) is gradual (Perdomo et al., 2015), hence in the existing literature, experiments usually adopts relatively long duration for drought stress. For instance, in a study evaluating drought stress effects on growth, yield, and physiological activities of rice varieties, the drought treatment duration was set to 14 days (Amin et al., 2022). A field investigation assessing seven rice cultivars under continuous irrigation regimes established drought exposures averaging 60 days to examine yield potential under water stress (Barnaby et al., 2019). Research on high temperature and water stress impacts during heading and grain filling stages implemented targeted 10-day drought treatments at heading phase to analyze pollen development and grain quality (Duan et al., 2012). An evaluation of long-term combined heat and water deficit stress on global crops imposed minimum 40-day drought treatments to quantify impacts on plant growth and water-use efficiency (Perdomo et al., 2015). What's more, the long-term stress (>20d) alters growth and water-use efficiency, no evidence confirms significant yield reduction from short-term stress (Costa et al., 2021).

The impact of short-duration drought on rice remains debated. While extremely severe but brief droughts

can be fatal, in some cases, rewatering after short-term stress can promote growth and increase biomass. During vegetative stages, drought enhances soil aeration and root-shoot ratio, improving nutrient/water uptake without compromising growth; while in reproductive stages, short-term drought triggers compensatory recovery post-stress, potentially accelerating grain filling without yield loss (Chi et al., 2001; Jiang et al., 2019; Li et al., 2005). To minimize the influence of these uncertain effects, we set a 10-day threshold to exclude short-duration drought events.

According to China's national standard for agricultural drought classification (Grade of agricultural drought-GB/T 32136-2015), relative soil moisture below 25% is classified as "extremely severe" drought. We analyzed the histogram of relative soil moisture in our study area and during the phenological stages of interest. The results showed that extremely severe drought events are relatively rare in frequency, with the cumulative frequency of single-day relative soil moisture $\leqslant$ 25% accounting for only 4% of the total histogram frequency (Figure R1). Therefore, we believe that excluding short-period droughts does not overlook a major portion of impactful drought events.

[Figure]

**Figure R1.** Histogram of single-day relative soil moisture during rice phenological stages across all stations. Bars with relative soil moisture $\leqslant$ 25% account for 4% of the total frequency.

Considering rice's recovery capacity, the hydrological buffering of paddy systems, and the low frequency of extremely severe drought, the 10-day threshold serves to filter out transient fluctuations while retaining events that pose a high risk of physiological disruption. We have incorporated this rationale, along with the explanation of the threshold-based continuous accumulation method, into the revised Methods section of the manuscript.

**RC1.2** Clarification of Kernel Density Estimate. Figures 2a and 2c are labeled as Kernel Density Estimates (KDEs), but the x-axis represents time (e.g., 1981–2018), which is not standard in KDE applications. It is confusing what variable is being smoothed, and how the density values

should be interpreted. If these are smoothed frequencies or rolling densities over time, the figure should be relabeled or revised accordingly. I recommend providing a more detailed explanation of the construction, including the variable used, kernel type, bandwidth selection, and the interpretation of density on a time axis.

**RE:** Sorry for the confusion caused by the Figures 2a and 2c. After careful reconsideration, we have decided to remove all KDE visualizations (Figs 2a/c). Instead, we have created separate bar plots for heat-drought (H1D1/H2D2/H3D3) and chilling-rainy (C2R2/C3R3) events (Figure R2). We have updated all relevant figures in the manuscript (Section 3.1) and supplemented the Methods section (2.5) with full implementation details.

We recognized a fundamental methodological mismatch in our initial approach. Applying KDEs directly to event occurrence years resulted in counterintuitive density interpretations along the time axis, as rightly highlighted by the reviewer. In addition, stacking multiple KDEs failed to resolve core visualization challenges. In our original construction:

- Variable Smoothed is the occurrence years of compound events (i.e., each year was a data point).

- Kernel Type: We used a Gaussian kernel.

- Bandwidth Selection: Bandwidth was selected automatically using Silverman's rule of thumb.

- Interpretation on Time Axis: The resulting density curve represented the *estimated probability density function of event occurrence across the years (1981-2018). Peaks indicated years with a higher relative concentration (density) of events, not higher frequency counts.

[Figure]

**Figure R2.** Absolute annual compound severity (CS) of concurrent heat-drought events during jointing-booting stage (a H1D1), heading-flowering (b H2D2), grain filling (c H3D3) for single-rice and concurrent chilling-rainy events during heading-flowering (d C2R2), grain filling (e C3R3) for late-rice for the period of 1981−2018. * indicates significant at the 0.05 Significance level.

**RC1.3** Interpretation and Modeling. The analysis relating yield anomalies to compound severity lacks clarity. Both axes in Figure 5 are restricted to negative values, with no explanation for this truncation. Are positive yield deviations and low-stress years excluded? If so, why?

**RE:** We had some reasons for the negative-axis constraint, but different for X-axis and Y-axis.

X-axis represents Compound severity, and its truncation stems from intrinsic metric properties. In our identification of events, we used the cumulative values surpassing certain threshold to compute severity (of either drought or heat stress), and applied copula fitting to derive joint exceedance probability density, based on which a standardized z-score were obtained to denote the compound severity. By definition, if temperature or moisture did not surpass their corresponding thresholds (as specified in Table 1 in the manuscript), severity will be 0. Correspondingly, we will have a truncation on the X-axis. To some extent, applying the threshold will exclude "low"-stress years, but those thresholds were obtained from national or local Standards, based on intensive field experiments.

Y-axis is about standardized Yield anomaly, derived from the detrended historical yield time series. In our previous version, we excluded positive yield anomalies, by assuming that years with compound climate extremes will strongly have negative yield impacts.

This design stems from the intrinsic properties of our metrics, the negative range of compound stress indices exclusively represents high-stress conditions, while negative anomalies directly measure loss magnitude. Positive values (reflecting favorable conditions, management optimizations, or uncaptured factors) were excluded as they represent distinct regimes, which could obscure the visual salience and scientific focus of the stress-loss relationship.

**RC1.4** Final Yield Model. Additionally, the use of simple scatterplots without formal statistical modeling is insufficient, given the complexity of the stress indices. I encourage the authors to fit and report a statistical model or clarify the final equation for this analysis to formally characterize the relationship between yield anomalies and compound stress severity. This would substantiate the visual patterns and improve analysis.

**RE:** Thank you! In direct response, we have decided to report a statistical model or clarify the final equation for this analysis to formally characterize the relationship between yield anomalies and compound stress severity, in both the method and result sections.

In the method section, we plan to add following information:

To reveal the statistical relationship between yield anomalies ($YA_t$) and compound severity ($CS$), simple linear regression analyses were conducted by using the equation below:

$$YA_t = \beta_0 + \beta_1 * CS + \varepsilon$$

where $YA_t$ is the standardized yield anomaly (detrended & normalized). $\beta_0$ is the intercept (expected yield anomaly at zero stress). $\beta_1$ is the yield loss per unit increase in compound severity). $\varepsilon$ is error term.

In the Methods section, we plan to report the fitting statistics and update the Figure 5 caption to report the fitted lines and the fitting statistics. This formal modeling substantiates the visual patterns and provides quantitative measures of stage-specific sensitivity of rice yield to compound climatic stress.

**RC1.5: Comments on Manuscript Structure and Flow**

Table 2 is referenced in the manuscript but not included.

**RE:** Sorry! This is a typo. It should be actually Table 1. Will revise.

The manuscript is generally well-organized, but there are several ways the narrative can be improved:

Abstract: Consider simplifying and using more intuitive phrasing to improve accessibility to the general scientific audience.

RE: Thank you for the detailed suggestions. We will follow your instructions to revise the manuscript and narrative, section by section.

**RE:** We appreciate the suggestion to improve accessibility. We will revise the Abstract to improve clarity and accessibility for a broader scientific readership.

Introduction: The rationale is well-motivated, but some repetition of literature gaps can be consolidated. Move technical details to Methods.

**RE:** We appreciate this point. To enhance clarity and narrative focus, we will consolidate repeated discussions on literature gaps in the Introduction. Moreover, technical content related to the definitions of compound events and their thresholds has been relocated to the Methods section (Section 2.3). These changes help streamline the Introduction and better emphasize the motivation, context, and scientific gaps addressed in this study.

Methods: While comprehensive, this section is very dense. I suggest creating a labeled subsections on "Compound Severity Metrics" that put together equations and definitions. A flowchart or schematic of the data-processing pipeline would improve readability.

**RE:** Thank you for the valuable suggestions to enhance readability. We try to implement the following changes:

We will create a dedicated subsection titled "2.4.1 Compound Severity Metrics". This subsection now clearly presents all relevant equations (e.g., the calculation of compound severity), definitions, and the rationale behind the chosen metrics.

We will add a new schematic diagram that visualizes the full data-processing workflow, from meteorological and phenological data input to compound severity assessment and yield impact analysis. This visual will aid improves reader understanding of the methodological framework.

Results: Avoid overuse of code-like labels (C2R2, H3D3) in narrative prose; use descriptive

names. Ensure all figures are introduced with clear interpretive framing.

**RE:** We agree that overusing code-like labels can hinder readability. Throughout the Results section (and the rest of the manuscript), labels such as H2D2, C3R3 will be replaced with descriptive names (e.g., " heat-drought events during heading-flowering stage#2 (H2D2)", "chilling-rainy events during grain filling stage#3 (C3R3)"). Furthermore, we have carefully reviewed the introduction of all figures. Each figure reference is now preceded by clear interpretive framing that explicitly states the scientific question or key finding the figure addresses (e.g., "To identify spatial hotspots of compound stress, Figure 3 shows..." or "Figure 5 reveals the relationship between compound stress severity and yield loss magnitude...").

Figures: Improve color bar labeling and add interpretive guidance in captions. Figures 3 and 5 in particular would benefit from better explanation of axis ranges and unit meanings.

**RE:** We thank the reviewer for the suggestions to improve figure clarity. We will revise the captions for Figures 3 and 5 as follows, incorporating enhanced color bar labeling and interpretive guidance:

Figure 3 revise caption:

"Spatial distribution of compound severity for concurrent heat-drought events in single-rice (a-c: jointing-booting#1, heading-flowering#2, grain filling stages#3) and concurrent chilling-rainy events in late-rice (d, e: heading-flowering#2, grain filling stages#3) during 1981−2018. Shading represents the magnitude of compound severity (unitless indices where more negative values indicate higher stress severity). Darker shades correspond to regions experiencing more intense compound stress."

Figure 5 revise caption:

"Relationship between compound stress severity and standardized yield anomaly (a-e: for specific growth stages and event types) and bar plot of standardized yield anomaly by growth stage (f) during 1995−2015. Axes are restricted to negative values to specifically focus on the relationship between yield loss magnitude (negative yield anomalies) and high compound stress severity (negative values) during damaging event years. The symbol * indicates F-test significance at the 10% level. Solid lines represent significant linear regression fits ($p < 0.01$) for stress-loss years (see Methods)."

These revised captions provide the necessary context for interpreting the figures, explicitly define the metrics and units, explain the axis ranges (especially for Fig 5), and offer guidance on how to interpret the visualizations.

Discussion: While informative, the discussion can be tightened.

**RE:** We plan to revise the Discussion section to improve conciseness and focus. Repetitive summaries of Results have been reduced. The section now more efficiently synthesizes key findings, emphasizes their novelty (especially regarding growth-stage-specific thresholds and impacts), places them clearly in the context of existing literature, and robustly discusses implications for adaptation and future research.

**References:**

Amin, M. W., Aryan, S., Habibi, N., Kakar, K., & Zahid, T. (2022). Elucidation of photosynthesis and yield performance of rice (Oryza sativa L.) under drought stress conditions. Plant Physiology Reports, 27(1), 143–151. https://doi.org/10.1007/s40502-021-00613-0

Barnaby, J. Y., Rohila, J. S., Henry, C. G., Sicher, R. C., Reddy, V. R., & McClung, A. M. (2019). Physiological and Metabolic Responses of Rice to Reduced Soil Moisture: Relationship of Water Stress Tolerance and Grain Production. International Journal of Molecular Sciences, 20(8), Article 8. https://doi.org/10.3390/ijms20081846

Chi D., Wang X., Zhu T., Xia  guimin, & Wang  wenyan. (2001). Water Saving and High Yield Irrigation Models of Rice and Soil Moisture Potential Control Criteria. Transactions of the Chinese Society of Agricultural Engineering, 17(2), 59–64.

Costa, M. V. J. D., Ramegowda, Y., Ramegowda, V., Karaba, N. N., Sreeman, S. M., & Udayakumar, M. (2021). Combined Drought and Heat Stress in Rice: Responses, Phenotyping and Strategies to Improve Tolerance. Rice Science, 28(3), Article 3. https://doi.org/10.1016/j.rsci.2021.04.003

Duan, H., Tang, Q., Ju, C., Liu, L., & Yang, J. (2012). Effect of High Temperature and Drought on Grain Yield and Quality of Different Rice Varieties During Heading and Early Grain Filling Periods. Scientia Agricultura Sinica, 45(22), 4561–4573. https://doi.org/10.3864/j.issn.0578-1752.2012.22.003

Jiang, Y., Carrijo, D., Huang, S., Chen, J., Balaine, N., Zhang, W., van Groenigen, K. J., & Linquist, B. (2019). Water management to mitigate the global warming potential of rice systems: A global meta-analysis. Field Crops Research, 234, 47–54. https://doi.org/10.1016/j.fcr.2019.02.010

Li, T., Li  shaohua, & Wang, J. (2005). Effects of water deficiency stress on transport and distribution of 14C-assimilates in micropropagated apple plants. Journal of China Agricultural University, 5, 44–48.

Perdomo, J. A., Conesa, M. À., Medrano, H., Ribas-Carbó, M., & Galmés, J. (2015). Effects of long-term individual and combined water and temperature stress on the growth of rice, wheat and maize: Relationship with morphological and physiological acclimation. Physiologia Plantarum, 155(2), Article 2. https://doi.org/10.1111/ppl.12303

---

## Author Comment (AC2)

**Title: Spatiotemporal variation of growth-stage specific concurrent climate extremes and their yield impacts for rice in southern China**

**Response to Reviewer Comments (RC2):**

**'Comment on egusphere-2025-1393', Anonymous Referee #2, 03 Jun 2025**

The paper has significantly improved compared to the earlier version. I thank the authors for taking the revision process seriously and applying the requested modifications.

In my view, the paper still requires more clarifications, particularly in the methods section and in how the results are contextualized within the broader literature:

**RE:** Thank you so much for your comments and suggestions on our manuscript. We will try to clarify the points that you have mentioned to improve the manuscript further. We have responded to the comments and suggestions point-by-point below (in blue).

**Major Comments:**

**RC2.1.1** Copulas are introduced but never mentioned in the results. Is the KDE introduced in Fig. 2 equivalent to the copula CDF? If so, the terminology needs to be harmonized. If the KDE represents something else, this should be clearly introduced in the methods section.

**RE:** Sorry for the confusing results. KDE is NOT equivalent to the copula CDF. The KDE figure tried to present the density of event occurrence along the time. As it confused both reviewers, we have decided to remove the KDE parts.

We offered the copula CDF results here for your reference. In the previous manuscript, copula CDF results were not presented directly, but the inverse-transformed exceedance probability of compound severity, derived directly from the copula CDF (Figure R1) of simultaneous exceedances of both climate variables above their growth-stage thresholds. These results were shown in Figure 2 (b, d), Figure 3, and Figure 5, where each map and time series embody the joint probability computed by the copula, converted to a standardized severity index via the inverse normal transform. However, per your suggestion, we have decided to supply copula CDF results in the supplementary material, so that readers could access the dependence pattern between each pair of two stress.

[Figure]

**Figure R1.** Copula cumulative distribution functions as 3D surface of u (heat severity) and v (drought severity) for concurrent heat-drought events during jointing-booting#1 (a, H1D1); heading-flowering#2 (b, H2D2); grain filling stages#3 (c, H3D3) and concurrent chilling-rainy events during heading-flowering#2 (d, C2R2); grain filling stages#3 (e, C3R3).

**RC2.1.2** In the copula section, the purpose of Lines 204–207 and Equation 6 is unclear. Isn't the joint probability (i.e., P(x > X, y > Y)) the main quantity of interest? If so, why not introduce Equation 7 directly? You may refer to this article for inspiration on copula methods and joint return periods: https://wires.onlinelibrary.wiley.com/doi/10.1002/wat2.1579.

**RE:** Thank you! Yes, the joint probability of (i.e., $P(X \leq x, Y \leq y)$) is the main quantity of interest. But due to our definition of severity for each individual stress, we have slightly modified the conventional formula to adapt to our case.

In our copula framework, Equation 5 implements the base copula function $C(u, v)$ as in the referenced literature's Equations 2 and 3 (Tootoonchi et al., 2022).

$$P(X \leq x, Y \leq y) = C[F(X), G(Y)] = C(u, v) \qquad (5)$$

In which $u$ and $v$ are the severity of individual stress, i.e. $S_{H1}$ and $S_{D1}$ for heat and drought in the joint-booting stage. According to our definition, our severity scores have many "0" values as in years that the threshold is not surpassed. Therefore, in the fitting process, samples that $u=0$ or $v=0$ were not included, and should be taken back into account when we derive the joint exceedance probability.

As our main quantity of interest is the joint exceedance probability $P(X > x, Y > y)$, we applied Equation 6:

$$P_{S_{H1}S_{D1}} = P(S_{H1} \geq x, S_{D1} \geq y | x > 0, y > 0) \cdot P(x > 0, y > 0) = [1 - u - v + C_{H1D1}(u, v)] \cdot \frac{n(x>0, y>0)}{N} \qquad (6)$$

Two calculations were included in this equation. We firstly converted exceedance probability by using

formula: $P(X > x, Y > y) = 1 - u - v + C_{H1D1}(u, v)$, where $u$ and $v$ are the marginal CDF values for each severity. Besides, we also applied the law of total probability through the conditional probability framework by multiplying the conditional exceedance probability $P(A|B)$ by the marginal event probability $P(B)$, yielding the overall joint probability $P(A)$. This transformation will get the years without compound events (either $u$=0 or $v$=0, not fitted in Equation (5)) back into account when computing the joint exceedance probability.

Finally, Equation 7 translates that joint probability into a severity index via the inverse transform, so that lower z-scores correspond to more severe compound extremes.

$$CS_{H1D1} = \varphi^{-1}[P_{S_{H_1}S_{D_1}}] \hspace{4cm} (7)$$

In the revision, we plan to expand Section 2.4 to clarify above issues, and to elaborate from raw copula CDF to joint exceedance probability, and then to normalized severity scores, with citations to (Li et al., 2022; Wu et al., 2021) and related copula literature.

**RC 2.2** Section 2.6 is rather generic. What are B1 and B2? Please introduce them properly. If B1 refers to climatic conditions and B2 to non-climatic factors, then from Line 416 onwards, a direct inference about the impact of infrastructure on yields cannot be made.

**RE:** Thank you for your question. Equation (9) is used to detrend historical yield time series to derive standardized yield anomalies, following the (Ye et al., 2015). In the equation, $\beta_0$ is the intercept, and $\beta_1$ is the slope of the regression line. $\beta_1$ captures the long-term exponential trend in yield improvement for which the literature generally assumes as technological trend (Holly Wang & Zhang, 2003). This formulation does not explicitly decompose climatic ($\beta_0$) and non-climatic ($\beta_1$) components.

Give above confusing situation, we plan to revise the text to clarify this equation, and its coefficients. The sentence from Line 416 and any related interpretation regarding non-climatic drivers like infrastructure will be removed to avoid unsupported inferences.

**RC 2.3** Discussion section: Please revise the text to reflect the broader implications of your findings and include only points that can be directly deduced from your analysis.

**RE:** Thank you for your guidance on tightening the Discussion. Our broader implication is of two folds: (1) While our study focuses on rice in southern China, the analytical framework is not crop- or region-specific, and may be applied to other major staple crops and agro-ecological zones; (2) The findings offer practical insights for managing compound extreme events in rice production systems in southern China. Speculative or unrelated content will be removed to ensure a clear and evidence-based narrative.

**RC 2.4 Specific Comments:**

L14: "Hamper" doesn't sound right.

RE: Thank you for the suggestion. We plan to replace it with "limit".

L116: Briefly introduce the two datasets at the end of this sentence before discussing them individually.

**RE:** Thank you for the suggestion. To improve flow, we will spell out the two datasets at the end of the first sentence: "We used two complementary rice phenology datasets: rice agrometeorological station observations dataset (1981–2018) (CMA, http://data.cma.cn) and the ChinaCropPhen1km dataset (2000-2019) (Luo et al., 2020)".

L121 (and repeated elsewhere, e.g., L163): What is "QX/T 468–2018"? This terminology is unclear. If it refers to internal coding, it may be unnecessary to mention.

**RE:** Thank you for the suggestion. "QX/T 468–2018" stands for Standard ("T") in the Meteorological Administration (QX stands for QiXiang, which is the Chinese pronunciation of Meteorology). "QX/T 468–2018" represents "Specifications for agrometeorological observation-Rice". We will explain the term and provide necessary information in the revision.

L248: Use "The impact of ⋯ on yield" instead of "yield impact."

**RE:** Thank you. We will revise as suggested.

Figure 4: I am not sure I understand what DC refers to. If it represents correlation, shouldn't the boxplot range be limited to 1? Why does it go up to 1.2 in panel d1 C2r2 for DCtot?

**RE:** In the path analyses, DC denotes the coefficient of determination derived from squared path coefficients ($DC_i = P_i^2$) and that the co-determination coefficient ($DC_{co}$) arises from the interaction term ($2P_i r_{ij} P_j$). Summing all direct and co-determination terms can yield a total $DC_{total}$ greater than 1, reflecting the combined explanatory power of individual and interactive effects. We will explicitly explain this so that readers understand why values may exceed unity. We will clarify in the revised Methods and figure caption.

L384–396: This section needs thorough revision. The reference to Zhang is problematic. Additionally, suggesting a dominant factor may not be valid, as these relationships are likely highly location- and case-specific. "Large" is not the right word here. Please remind the reader what "#3" refers to.

**RE:** Thank you so much for the suggestion. We will rewrite lines 384–396 to remove the problematic Zhang citation, avoid implying any universally dominant driver, and replace "large" with more precise descriptors. We will also clarify that "#3" refers to the grain-filling stage. The revised text will focus strictly on our own stage-specific findings without overgeneralization.

L417: Use "Different impacts of ⋯ on yields" instead of "yield impacts."

**RE:** Thank you. We will revise as suggested.

L421: Were these losses shown in any figures or derived from your analysis? If not, consider removing this sentence. Also, since the study does not directly assess the impact of irrigation, that discussion may not be relevant.

**RE:** Thank you. The losses don't come from my results. We will delete this sentence and the related discussion of irrigation, ensuring our narrative remains confined to results derived directly from our analysis.

L437: Replace "rainy stress" with "rain stress."

**RE:** Thank you. We will revise as suggested.

L456: On what plots are these spatial shifts in concurrent events shown? If you refer to shifts over time, clarify this. If not, the sentence is unclear.

RE: We acknowledge the confusion. The term "shifted" misleadingly suggests a temporal change; in fact, we intended only to describe the changes in the compound heat-drought hotspots by rice growth stage. We will rephrase this passage (in both the main text and abstract) to clearly convey that these are spatial distribution characteristics, not temporal shifts.

L457: "Spatial difference in phenology" is unclear, please rephrase.

**RE:** We will rephrase this sentence to: "These spatial patterns are driven primarily by differences in crop phenology across locations—such as the timing of flowering in early versus late rice, rather than by the spatial distribution of extreme climate conditions."

L463 onwards (Conclusion): The conclusion is not the right place to introduce new references or discuss limitations. Consider revising this section and relocating these points to more appropriate sections in the manuscript.

**RE:** We will confine the Conclusion to summarizing key findings already presented and remove any newly cited literature or discussions of limitations. All material on study constraints and future directions will be moved to the Discussion or Methods as appropriate.

**References:**

Holly Wang, H., & Zhang, H. (2003). On the Possibility of a Private Crop Insurance Market: A Spatial Statistics Approach. *Journal of Risk and Insurance*, *70*(1), 111–124. https://doi.org/10.1111/1539-6975.00051

Li, Z., Liu, W., Ye, T., Chen, S., & Shan, H. (2022). Observed and CMIP6 simulated occurrence and intensity of compound agroclimatic extremes over maize harvested areas in China. *Weather and Climate Extremes*, *38*, 100503. https://doi.org/10.1016/j.wace.2022.100503

Luo, Y., Zhang, Z., Chen, Y., Li, Z., & Tao, F. (2020). ChinaCropPhen1km: A high-resolution crop phenological dataset for three staple crops in China during 2000–2015 based on leaf area index (LAI) products. *Earth System Science Data*, *12*(1), 197–214. https://doi.org/10.5194/essd-12-197-2020

Tootoonchi, F., Sadegh, M., Haerter, J. O., Räty, O., Grabs, T., & Teutschbein, C. (2022). Copulas for hydroclimatic analysis: A practice-oriented overview. *WIREs Water*, *9*(2), e1579. https://doi.org/10.1002/wat2.1579

Wu, H., Su, X., & Singh, V. P. (2021). Blended Dry and Hot Events Index for Monitoring Dry-Hot Events

Over Global Land Areas. *Geophysical Research Letters*, *48*(24), e2021GL096181. https://doi.org/10.1029/2021GL096181

Ye, T., Nie, J., Wang, J., Shi, P., & Wang, Z. (2015). Performance of detrending models of crop yield risk assessment: Evaluation on real and hypothetical yield data. *Stochastic Environmental Research and Risk Assessment*, *29*(1), 109–117. https://doi.org/10.1007/s00477-014-0871-x

---

## Author Response (AR1)

- 1 Title: Spatiotemporal variation of growth-stage specific concurrent climate
- 2 extremes and their yield impacts for rice in southern China
- 3 Response to Reviewer Comments (RC1):
- 4 'Comment on egusphere-2025-1393', Anonymous Referee #1, 20 May 2025
- 5 The manuscript presents a well-designed and timely study on the correlation between
- 6 compound climate extremes and rice yields in southern China, with clear relevance to
- 7 climate change adaptation. The authors leverage growth-stage-specific physiological
- 8 thresholds, multi-source gridded data, and compound severity metrics to offer new
- 9 insights into how concurrent heat-drought and chilling-rainy events affect rice
- production. This work makes an important contribution in the construction of metrics
- 11 for compound stressors. However, several points require clarification, and
- improvements in structure and presentation would significantly improve the manuscript.
- 13 **RE:** Thank you so much for your comments and suggestions on our manuscript. We have
- responded to the comments and suggestions point-by-point below (in blue).

**Major Comments:**

- 16 **RC1.1** Ambiguity in Drought Stress Severity Definition. The calculation of drought
- severity appears to exclude events shorter than 10 days, regardless of intensity. Please
- clarify whether severity is accumulated continuously or only calculated if a 10-day
- event threshold is met. For rice, a very low soil moisture period, even for a week, can
- 20 be fatal. Justification for this duration cutoff should be provided, ideally based on
- 21 physiological or agronomic evidence.
- 22 **RE:** Thank you for the question. The calculation of drought severity accumulates from Day 1
- 23 (the onset of soil moisture falling below the defined threshold). However, we retain and analyze
- only events persisting for  $\geq 10$  consecutive days. The threshold of ten days was applied based
- on physiological and agronomic relevance and experimental evidence.
- 26 Drought development in field environments (especially paddies) is gradual (Perdomo et al.,
- 27 2015), hence in the existing literature, experiments usually adopts relatively long duration for
- 28 drought stress. For instance, in a study evaluating drought stress effects on growth, yield, and
- 29 physiological activities of rice varieties, the drought treatment duration was set to 14 days
- 30 (Amin et al., 2022). A field investigation assessing seven rice cultivars under continuous
- 31 irrigation regimes established drought exposures averaging 60 days to examine yield potential
- 32 under water stress (Barnaby et al., 2019). Research on high temperature and water stress
- 33 impacts during heading and grain filling stages implemented targeted 10-day drought

treatments at heading phase to analyze pollen development and grain quality (Duan et al., 2012). An evaluation of long-term combined heat and water deficit stress on global crops imposed minimum 40-day drought treatments to quantify impacts on plant growth and water-use efficiency (Perdomo et al., 2015). What's more, the long-term stress (>20d) alters growth and water-use efficiency, no evidence confirms significant yield reduction from short-term stress (Costa et al., 2021).

The impact of short-duration drought on rice remains debated. While extremely severe but brief droughts can be fatal, in some cases, rewatering after short-term stress can promote growth and increase biomass. During vegetative stages, drought enhances soil aeration and root-shoot ratio, improving nutrient/water uptake without compromising growth; while in reproductive stages, short-term drought triggers compensatory recovery post-stress, potentially accelerating grain filling without yield loss (Chi et al., 2001; Jiang et al., 2019; Li et al., 2005). To minimize the influence of these uncertain effects, we set a 10-day threshold to exclude short-duration drought events.

According to China's national standard for agricultural drought classification (Grade of agricultural drought-GB/T 32136-2015), relative soil moisture below 25% is classified as "extremely severe" drought. We analyzed the histogram of relative soil moisture in our study area and during the phenological stages of interest. The results showed that extremely severe drought events are relatively rare in frequency, with the cumulative frequency of single-day relative soil moisture ≤25% accounting for only 4% of the total histogram frequency (Figure R1). Therefore, we believe that excluding short-period droughts does not overlook a major portion of impactful drought events.

**Figure R1.** Histogram of single-day relative soil moisture during rice phenological stages across all stations. Bars with relative soil moisture ≤25% account for 4% of the total frequency.

- 59 Considering rice's recovery capacity, the hydrological buffering of paddy systems, and the low
- frequency of extremely severe drought, the 10-day threshold serves to filter out transient
- fluctuations while retaining events that pose a high risk of physiological disruption. We have
- 62 incorporated this rationale, along with the explanation of the threshold-based continuous
- accumulation method, into the revised Methods section of the manuscript, please refer to Lines
- 64 178-185.
- 65 RC1.2 Clarification of Kernel Density Estimate. Figures 2a and 2c are labeled as
- 66 Kernel Density Estimates (KDEs), but the x-axis represents time (e.g., 1981–2018),
- 67 which is not standard in KDE applications. It is confusing what variable is being
- smoothed, and how the density values should be interpreted. If these are smoothed
- frequencies or rolling densities over time, the figure should be relabeled or revised
- accordingly. I recommend providing a more detailed explanation of the construction,
- including the variable used, kernel type, bandwidth selection, and the interpretation of
- density on a time axis.
- 73 **RE:** Sorry for the confusion caused by the Figures 2a and 2c. After careful reconsideration, we
- 74 have decided to remove all KDE visualizations (Figs 2a/c). Instead, we have created separate
- bar plots for heat-drought (H1D1/H2D2/H3D3) and chilling-rain (C2R2/C3R3) events (Figure
- R2). We have updated relevant figure and results in the manuscript (Section 3.1, Lines 298-
- 77 313).
- We recognized a fundamental methodological mismatch in our initial approach. Applying
- 79 KDEs directly to event occurrence years resulted in counterintuitive density interpretations
- along the time axis, as rightly highlighted by the reviewer. In addition, stacking multiple KDEs
- 81 failed to resolve core visualization challenges. In our original construction:
- 82 Variable Smoothed is the occurrence years of compound events (i.e., each year was a data
- 83 point).
- Kernel Type: We used a Gaussian kernel.
- 85 Bandwidth Selection: Bandwidth was selected automatically using Silverman's rule of
- 86 thumb.
- 87 Interpretation on Time Axis: The resulting density curve represented the \*estimated
- probability density function of event occurrence across the years (1981-2018). Peaks
- 89 indicated years with a higher relative concentration (density) of events, not higher
- 90 frequency counts.

Figure R2. Annual compound severity of concurrent compound events during 1981–2018.

Panels (a–c) show the concurrent heat–drought events in single–rice during jointing–booting#1 (H1D1), heading–flowering#2 (H2D2), grain filling stages#3 (H3D3). Panels (d–e) show the concurrent chilling–rain events in late–rice during heading–flowering#2 (C2R2), grain filling stages#3 (C3R3). \* and \*\* indicate statistically significant at the significance levels of 0.05 and 0.01, respectively.

**RC1.3** Interpretation and Modeling. The analysis relating yield anomalies to compound severity lacks clarity. Both axes in Figure 5 are restricted to negative values, with no explanation for this truncation. Are positive yield deviations and low-stress years excluded? If so, why?

**RE:** We applied the negative-axis constraint for specific reasons, which differ between the X-axis and the Y-axis.

X-axis represents Compound severity, and its truncation stems from intrinsic metric properties.

In our identification of events, we used the cumulative values surpassing certain threshold to

106 compute severity, and applied copula fitting to derive joint exceedance probability density,

based on which a standardized z-score were obtained to denote the compound severity. By the

definition, if temperature or moisture did not surpass their corresponding thresholds (as

specified in Table 1 in the manuscript), severity would be 0. Correspondingly, we will have a

truncation on the X-axis. To some extent, applying the threshold will exclude "low"-stress years,

but those thresholds were obtained from national or local Standards, based on intensive field

112 experiments.

109

110

- Y-axis is about standardized Yield anomaly, derived from the detrended historical yield time
- series. In our previous version, we excluded positive yield anomalies, by assuming that years
- with compound climate extremes will strongly have negative yield impacts.
- This design stems from the intrinsic properties of our metrics, the negative range of compound
- stress indices exclusively represents high-stress conditions, while negative anomalies directly
- measure loss magnitude. Positive values (reflecting favorable conditions, management
- optimizations, or uncaptured factors) were excluded as they represent distinct regimes, which
- could obscure the visual salience and scientific focus of the stress-loss relationship.
- To clarify, we have added corresponding descriptions in the Methods section 2.6 (Lines 295-
- 226), Results section 3.4 (Lines 374-375), and the caption of Figure 5 (Line 393).
- 123 **RC1.4** Final Yield Model. Additionally, the use of simple scatterplots without formal
- 124 statistical modeling is insufficient, given the complexity of the stress indices. I
- encourage the authors to fit and report a statistical model or clarify the final equation
- for this analysis to formally characterize the relationship between yield anomalies and
- 127 compound stress severity. This would substantiate the visual patterns and improve
- 128 analysis.
- 129 **RE:** Thank you! In direct response, we have reported a statistical model or clarify the final
- 130 equation for this analysis to formally characterize the relationship between yield anomalies and
- compound stress severity, in both the method and result sections.
- 132 In the method section (Lines 289-296), we added following information (simplified here):
- To reveal the statistical relationship between yield anomalies  $(YA_t)$  and compound severity
- 134 (CS), simple linear regression analyses were conducted by using the equation below:
- 135  $YA_t = \beta_0 + \beta_1 * CS + \varepsilon$
- where  $YA_t$  is the standardized yield anomaly (detrended & normalized).  $\beta_0$  is the intercept
- 137 (expected yield anomaly at zero stress).  $\beta_1$  is the yield loss per unit increase in compound

- 138 severity).  $\varepsilon$  is error term.
- In the Results section 3.4 (Lines 369-387), we reported the fitting statistics and update the
- Figure 5 caption (Lines 388-394) to report the fitted lines and the fitting statistics. This formal
- modeling substantiated the visual patterns and provided quantitative measures of stage-specific
- sensitivity of rice yield to compound climatic stress.

**RC1.5: Comments on Manuscript Structure and Flow**

- Table 2 is referenced in the manuscript but not included.
- 145 **RE:** Sorry! This is a typo. We have thoroughly rechecked the table labels in the manuscript,
- and confirmed that they were now correct.
- The manuscript is generally well-organized, but there are several ways the narrative
- can be improved:

- 149 Abstract: Consider simplifying and using more intuitive phrasing to improve
- accessibility to the general scientific audience.
- 151 **RE:** We appreciate the suggestion to improve accessibility. We have revised the Abstract (Lines
- 152 13-31) to improve clarity and accessibility for a broader scientific readership.
- 153 Introduction: The rationale is well-motivated, but some repetition of literature gaps can
- be consolidated. Move technical details to Methods.
- 155 **RE:** We appreciate this point. To enhance clarity and narrative focus, we have simplified
- repeated discussions on literature gaps in the Introduction by summarizing the literature gaps
- in a single section (Lines 56–66) and avoiding repeated mentions in other paragraphs (Lines
- 49–55). Moreover, technical content related to the stress types and their thresholds (Lines 81-
- 159 89) has been relocated to the Methods section 2.3. These changes helped streamline the
- 160 Introduction and better emphasize the motivation, context, and scientific gaps addressed in this
- 161 study.
- Methods: While comprehensive, this section is very dense. I suggest creating a labeled
- subsections on "Compound Severity Metrics" that put together equations and
- definitions. A flowchart or schematic of the data-processing pipeline would improve
- 165 readability.
- 166 **RE:** Thank you for the valuable suggestions. To enhance readability, the following changes
- 167 have been implemented:
- To improve clarity and avoid excessive density in this section, we have reorganized the content

- into two main sections (Lines 147-244): 2.3 Individual Extreme Types and Severity Metrics
- 170 (Line 147) and 2.4 Compound Climate Extreme Types and Severity Metrics (Line 191).
- Within each section, we first describe the respective event types and threshold (2.3.1 Individual
- 172 Extremes Considered and 2.4.1 Compound Climate Extreme Types), followed by a newly
- added subsection detailing the methods for calculating severity (2.3.2 Severity Metrics for
- 174 Individual Events and 2.4.2 Compound severity metric).
- 175 Results: Avoid overuse of code-like labels (C2R2, H3D3) in narrative prose; use
- descriptive names. Ensure all figures are introduced with clear interpretive framing.
- 177 **RE:** We agree that overusing code-like labels can hinder readability. Throughout the Results
- section (and the rest of the manuscript), labels such as H2D2, C3R3 will be replaced with
- descriptive names (e.g., " heat-drought events during heading-flowering stage#2 (H2D2)",
- "chilling-rain events during grain filling stage#3 (C3R3)" (Lines 302-303, 306-307, 320-322,
- 181 333-335, 376...). Furthermore, we have carefully reviewed the introduction of all figures. Each
- figure reference is now preceded by clear interpretive framing that explicitly states the scientific
- question or key finding the figure addresses (e.g., " *Specifically, the annual compound severity*
- 184 for each type of concurrent climate extremes was averaged within each grid cell to identify and
- map spatial hotspots (Fig. 3) "(Lines 316-317) or " Figure 5 presents the fitted data points and
- the regression trend lines to visually illustrate the models." (Line 373).
- 187 Figures: Improve color bar labeling and add interpretive guidance in captions. Figures
- 3 and 5 in particular would benefit from better explanation of axis ranges and unit
- meanings.
- 190 **RE:** We thank the reviewer for the suggestions to improve figure clarity. We have revised all
- 191 figures accordingly, particularly Figures 3 and 5, including improved color bar labeling and
- added interpretive guidance.
- 193 Figure 3 revise caption (Lines 327-331):
- 194 "Figure 3. Spatial distribution of compound severity for concurrent climate extremes during
- 195 1981–2018. Panels (a–c) show concurrent heat–drought events in single–rice during jointing–
- booting#1 (H1D1), heading\_flowering#2 (H2D2), and grain filling stages#3 (H3D3). Panels
- 197 (d—e) show concurrent chilling—rain events in late—rice during heading—flowering#2 (C2R2),
- and grain filling stages#3 (C3R3). Shading represents compound severity (unitless index), with
- 199 darker colors indicating higher stress severity. "
- Figure 5 revise caption (Lines 389-392):
- 201 "Figure 5. Relationship between compound severity and standardized yield anomaly during

- 202 1995–2015. Panels (a–c) show concurrent heat–drought events for single–rice during jointing–
- booting#1 (H1D1), heading-flowering#2 (H2D2), grain filling stages#3 (H3D3). Panels (d-e)
- show concurrent chilling—rain events for late—rice during heading—flowering#2 (C2R2), grain
- 205 filling stages#3 (C3R3). \*\*\* indicates statistically significant at the significance levels of
- 206 *0.001*."
- 207 These revised captions provided the necessary context for interpreting the figures, explicitly
- define the metrics and units, explain the axis ranges (especially for Fig 5), and offer guidance
- 209 on how to interpret the visualizations.
- 210 Discussion: While informative, the discussion can be tightened.
- 211 RE: We have revised the Discussion section to improve conciseness and focus. Repetitive
- summaries of Results have been reduced. In particular, section 4.2 (Lines 431-449) and 4.3
- 213 (Lines 463-479) have been streamlined to clearly present the key findings, highlight their
- 214 novelty (especially regarding growth-stage-specific thresholds and impacts), place them in the
- 215 context of existing literature, and discuss their implications for adaptation and future research
- in greater depth.

**217 **References:**

- 218 Amin, M. W., Aryan, S., Habibi, N., Kakar, K., & Zahid, T. (2022). Elucidation of
- 219 photosynthesis and yield performance of rice (Oryza sativa L.) under drought stress
- 220 conditions. Plant Physiology Reports, 27(1), 143–151. https://doi.org/10.1007/s40502-
- 221 021-00613-0
- 222 Barnaby, J. Y., Rohila, J. S., Henry, C. G., Sicher, R. C., Reddy, V. R., & McClung, A. M. (2019).
- 223 Physiological and Metabolic Responses of Rice to Reduced Soil Moisture: Relationship
- of Water Stress Tolerance and Grain Production. International Journal of Molecular
- 225 Sciences, 20(8), Article 8. https://doi.org/10.3390/ijms20081846
- 226 Chi D., Wang X., Zhu T., Xia guimin, & Wang wenyan. (2001). Water Saving and High
- 227 Yield Irrigation Models of Rice and Soil Moisture Potential Control Criteria. Transactions
- of the Chinese Society of Agricultural Engineering, 17(2), 59–64.
- 229 Costa, M. V. J. D., Ramegowda, Y., Ramegowda, V., Karaba, N. N., Sreeman, S. M., &
- Udayakumar, M. (2021). Combined Drought and Heat Stress in Rice: Responses,
- 231 Phenotyping and Strategies to Improve Tolerance. Rice Science, 28(3), Article 3.
- 232 https://doi.org/10.1016/j.rsci.2021.04.003
- Duan, H., Tang, Q., Ju, C., Liu, L., & Yang, J. (2012). Effect of High Temperature and Drought
- on Grain Yield and Quality of Different Rice Varieties During Heading and Early Grain

- 235 Filling Periods. Scientia Agricultura Sinica, 45(22), 4561–4573.
- 236 https://doi.org/10.3864/j.issn.0578-1752.2012.22.003
- Jiang, Y., Carrijo, D., Huang, S., Chen, J., Balaine, N., Zhang, W., van Groenigen, K. J., &
- Linquist, B. (2019). Water management to mitigate the global warming potential of rice
- systems: A global meta-analysis. Field Crops Research, 234, 47–54.
- 240 https://doi.org/10.1016/j.fcr.2019.02.010
- 241 Li, T., Li, shaohua, & Wang, J. (2005). Effects of water deficiency stress on transport and
- distribution of 14C-assimilates in micropropagated apple plants. Journal of China
- 243 Agricultural University, 5, 44–48.
- Perdomo, J. A., Conesa, M. A., Medrano, H., Ribas-Carbó, M., & Galmés, J. (2015). Effects of
- long-term individual and combined water and temperature stress on the growth of rice,
- wheat and maize: Relationship with morphological and physiological acclimation.
- 247 Physiologia Plantarum, 155(2), Article 2. https://doi.org/10.1111/ppl.12303
- 248 Title: Spatiotemporal variation of growth-stage specific concurrent climate
- 249 extremes and their yield impacts for rice in southern China
- 250 Response to Reviewer Comments (RC2):
- 251 'Comment on equsphere-2025-1393', Anonymous Referee #2, 03 Jun 2025
- 252 The paper has significantly improved compared to the earlier version. I thank the
- 253 authors for taking the revision process seriously and applying the requested
- 254 modifications.
- In my view, the paper still requires more clarifications, particularly in the methods
- section and in how the results are contextualized within the broader literature:
- 257 **RE:** Thank you so much for your comments and suggestions on our manuscript. We have
- 258 clarified the points that you have mentioned to improve the manuscript further. We have
- responded to the comments and suggestions point-by-point below (in blue).
- 260 **Major Comments:**
- 261 RC2.1.1 Copulas are introduced but never mentioned in the results. Is the KDE
- introduced in Fig. 2 equivalent to the copula CDF? If so, the terminology needs to be
- 263 harmonized. If the KDE represents something else, this should be clearly introduced
- in the methods section.
- 265 **RE:** Sorry for the confusing results. KDE is NOT equivalent to the copula CDF. The KDE

figures tried to present the density of event occurrence along the time. As it has confused both reviewers, we have decided to remove the KDE parts (Line 309).

We offered the copula CDF results here for your reference. In the previous manuscript, copula CDF results were not presented directly, but the inverse-transformed exceedance probability of compound severity, derived directly from the copula CDF (Figure R1) of simultaneous exceedances of both climate variables above their growth-stage thresholds. These results were shown in Figure 2 (b, d), Figure 3, and Figure 5, where each map and time series embody the joint probability computed by the copula, converted to a standardized severity index via the inverse normal transform. In the revision, we have explained the figure carefully, and supplied the copula CDFs in the Appendix A: Additional Figures section (Fig. A1, Lines 526-530).

**Figure R1.** Copula cumulative distribution functions as 3D surface of u (heat or chilling severity) and v (drought or rain severity) for concurrent heat-drought events during jointing-booting#1 (a, H1D1); heading-flowering#2 (b, H2D2); grain filling stages#3 (c, H3D3) and concurrent chilling-rain events during heading-flowering#2 (d, C2R2); grain filling stages#3 (e, C3R3).

**RC2.1.2** In the copula section, the purpose of Lines 204–207 and Equation 6 is unclear. Isn't the joint probability (i.e., P(x > X, y > Y)) the main quantity of interest? If so, why not introduce Equation 7 directly? You may refer to this article for inspiration on copula methods and joint return periods: https://wires.onlinelibrary.wiley.com/doi/10.1002/wat2.1579.

- **RE:** Thank you! Yes, the joint probability of (i.e.,  $P(X \le x, Y \le y)$ ) is the main quantity of
- 288 interest. But due to our definition of severity for each individual stress, we have slightly
- 289 modified the conventional formula to adapt to our case.
- In our copula framework, Equation 5 implements the base copula function C(u, v) as in the
- referenced literature's Equations 2 and 3 (Tootoonchi et al., 2022).
- 292  $P(X \le x, Y \le y) = C[F(X), G(Y)] = C(u, v)$  (5)
- In which u and v are the severity of individual stress, i.e.  $S_{H1}$  and  $S_{D1}$  for heat and drought in
- 294 the joint-booting stage. According to our definition, our severity scores have many "0" values
- as in years that the threshold is not surpassed. Therefore, in the fitting process, samples that u=0
- or v=0 were not included, and should be taken back into account when we derive the joint
- 297 exceedance probability.
- 298 As our main quantity of interest is the joint exceedance probability P(X > x, Y > y), we
- 299 applied Equation 6:
- 300  $P_{S_{H1}S_{D1}} = P(S_{H1} \ge x, S_{D1} \ge y | x > 0, y > 0) \cdot P(x > 0, y > 0) = [1 u v + C_{H1D1}(u, v)] \cdot P(x > 0, y > 0) = [1 u v + C_{H1D1}(u, v)] \cdot P(x > 0, y > 0) = [1 u v + C_{H1D1}(u, v)] \cdot P(x > 0, y > 0) = [1 u v + C_{H1D1}(u, v)] \cdot P(x > 0, y > 0) = [1 u v + C_{H1D1}(u, v)] \cdot P(x > 0, y > 0) = [1 u v + C_{H1D1}(u, v)] \cdot P(x > 0, y > 0) = [1 u v + C_{H1D1}(u, v)] \cdot P(x > 0, y > 0) = [1 u v + C_{H1D1}(u, v)] \cdot P(x > 0, y > 0) = [1 u v + C_{H1D1}(u, v)] \cdot P(x > 0, y > 0) = [1 u v + C_{H1D1}(u, v)] \cdot P(x > 0, y > 0) = [1 u v + C_{H1D1}(u, v)] \cdot P(x > 0, y > 0) = [1 u v + C_{H1D1}(u, v)] \cdot P(x > 0, y > 0) = [1 u v + C_{H1D1}(u, v)] \cdot P(x > 0, y > 0) = [1 u v + C_{H1D1}(u, v)] \cdot P(x > 0, y > 0) = [1 u v + C_{H1D1}(u, v)] \cdot P(x > 0, y > 0) = [1 u v + C_{H1D1}(u, v)] \cdot P(x > 0, y > 0) = [1 u v + C_{H1D1}(u, v)] \cdot P(x > 0, y > 0) = [1 u v + C_{H1D1}(u, v)] \cdot P(x > 0, y > 0) = [1 u v + C_{H1D1}(u, v)] \cdot P(x > 0, y > 0) = [1 u v + C_{H1D1}(u, v)] \cdot P(x > 0, y > 0) = [1 u v + C_{H1D1}(u, v)] \cdot P(x > 0, y > 0) = [1 u v + C_{H1D1}(u, v)] \cdot P(x > 0, y > 0) = [1 u v + C_{H1D1}(u, v)] \cdot P(x > 0, y > 0) = [1 u v + C_{H1D1}(u, v)] \cdot P(x > 0, y > 0) = [1 u v + C_{H1D1}(u, v)] \cdot P(x > 0, y > 0) = [1 u v + C_{H1D1}(u, v)] \cdot P(x > 0, y > 0) = [1 u v + C_{H1D1}(u, v)] \cdot P(x > 0, y > 0) = [1 u v + C_{H1D1}(u, v)] \cdot P(x > 0, y > 0) = [1 u v + C_{H1D1}(u, v)] \cdot P(x > 0, y > 0) = [1 u v + C_{H1D1}(u, v)] \cdot P(x > 0, y > 0) = [1 u v + C_{H1D1}(u, v)] \cdot P(x > 0, y > 0) = [1 u v + C_{H1D1}(u, v)] \cdot P(x > 0, y > 0) = [1 u v + C_{H1D1}(u, v)] \cdot P(x > 0, y > 0) = [1 u v + C_{H1D1}(u, v)] \cdot P(x > 0, y > 0) = [1 u v + C_{H1D1}(u, v)] \cdot P(x > 0, y > 0) = [1 u v + C_{H1D1}(u, v)] \cdot P(x > 0, y > 0) = [1 u v + C_{H1D1}(u, v)] \cdot P(x > 0, y > 0) = [1 u v + C_{H1D1}(u, v)] \cdot P(x > 0, y > 0) = [1 u v + C_{H1D1}(u, v)] \cdot P(x > 0, y > 0) = [1 u v + C_{H1D1}(u, v)] \cdot P(x > 0, y > 0) =$
- 301  $\frac{n(x>0,y>0)}{N}$  (6)
- 302 Two calculations were included in this equation. We firstly converted exceedance probability
- 303 by using formula:  $P(X > x, Y > y) = 1 u v + C_{H1D1}(u, v)$ , where u and v are the
- marginal CDF values for each severity. Besides, we also applied the law of total probability
- 305 through the conditional probability framework by multiplying the conditional exceedance
- 306 probability P(A|B) by the marginal event probability P(B), yielding the overall joint
- probability P(A). This transformation will get the years without compound events (either u=0
- 308 or v=0, not fitted in Equation (5)) back into account when computing the joint exceedance
- 309 probability.
- 310 Finally, Equation 7 translates that joint probability into a severity index via the inverse
- 311 transform, so that lower z-scores correspond to more severe compound extremes.
- 312  $CS_{H1D1} = \varphi^{-1}[P_{S_{H1}S_{D1}}]$  (7)
- 313 We have re-organized section 2.4.2 (Lines 202-244) to clarify above issues, and to provide a
- detailed explanation of the process, from marginal and joint modeling using copulas to the joint
- 315 exceedance probability, and then to the normalized severity scores
- RC 2.2 Section 2.6 is rather generic. What are B1 and B2? Please introduce them
- 317 properly. If B1 refers to climatic conditions and B2 to non-climatic factors, then from
- Line 416 onwards, a direct inference about the impact of infrastructure on yields cannot
- 319 be made.

- 320 **RE:** Thank you for your question. Equation (9) (revised Eq.10) is used to detrend historical
- 321 yield time series to derive standardized yield anomalies, following (Ye et al., 2015). In the
- equation,  $\beta_0$  is the intercept, and  $\beta_1$  is the slope of the regression line.  $\beta_1$  captures the long-
- 323 term exponential trend in yield improvement for which the literature generally assumes as
- 324 technological trend (Holly Wang & Zhang, 2003). This formulation does not explicitly
- decompose climatic  $(\beta_0)$  and non-climatic  $(\beta_1)$  components.
- 326 Give above confusing situation, we have revised the text to clarify this equation and its
- 327 coefficients (Lines 276-288). The descriptions in lines 473 and 477 regarding non-climatic
- 328 drivers (such as infrastructure) were used to explore possible reasons for the severe yield
- reductions in the Sichuan region and had no relation to Equation 10.
- 330 RC 2.3 Discussion section: Please revise the text to reflect the broader implications of
- your findings and include only points that can be directly deduced from your analysis.
- 332 **RE:** Thank you for your guidance on tightening the Discussion. Our broader implication is of
- two folds: (1) While our study focuses on rice in southern China, the analytical framework is
- not crop- or region-specific, and may be applied to other major staple crops and agro-ecological
- zones; (2) The findings offer practical insights for managing compound extreme events in rice
- production systems in southern China.

**RC 2.4 Specific Comments:**

- 338 L14: "Hamper" doesn't sound right.
- RE: Thank you for the suggestion. We have replaced it with "limit" (Line 14).
- L116: Briefly introduce the two datasets at the end of this sentence before discussing
- 341 them individually.

- **RE:** Thank you for the suggestion. To improve flow, we have spelt out the two datasets at the
- end of the first sentence (Line 111-112): "We used two complementary rice phenology datasets:
- rice agrometeorological station observations dataset (1981–2018) (CMA, <a href="http://data.cma.cn">http://data.cma.cn</a>)
- and the ChinaCropPhen1km dataset (2000-2019) (Luo et al., 2020)".
- 346 L121 (and repeated elsewhere, e.g., L163): What is "QX/T 468-2018"? This
- 347 terminology is unclear. If it refers to internal coding, it may be unnecessary to
- 348 mention.
- 349 **RE:** Thank you for the suggestion. "QX/T 468-2018" stands for Standard ("T") in the
- 350 Meteorological Administration (QX stands for QiXiang, which is the Chinese pronunciation of
- 351 Meteorology). "QX/T 468-2018" represents "Specifications for agrometeorological

- observation-Rice" developed in 2018. We have removed the code and explained the term,
- providing necessary information in the revision (Lines 117-118). However, the codes cited in
- lines 161-164 are publicly available standard numbers. As many similar standards exist, we
- 355 have retained these codes so that readers can clearly trace the exact standards referenced in our
- 356 study.
- 357 L248: Use "The impact of ... on yield" instead of "yield impact."
- 358 **RE:** Thank you. We have revised as suggested (Line 276) and have revised all similar
- statements throughout the manuscript accordingly (Lines 2, 25, 28, 82, 276, 369, 463, 468,
- 360 470, and 500).
- Figure 4: I am not sure I understand what DC refers to. If it represents correlation,
- shouldn't the boxplot range be limited to 1? Why does it go up to 1.2 in panel d1 C2r2
- 363 for DCtot?
- 364 **RE:** In the path analyses, DC denotes the coefficient of determination derived from squared
- path coefficients ( $DC_i = P_i^2$ ) and that the co-determination coefficient ( $DC_{co}$ ) arises from the
- interaction term  $(2P_ir_{ij}P_i)$ . Summing all direct and co-determination terms can yield a total
- 367  $DC_{total}$  greater than 1, reflecting the combined explanatory power of individual and
- 368 interactive effects.
- 369 In the revision, we have explicitly explained this so that readers understand why values may
- exceed unity. We have clarified in the revised Methods 2.5 (Lines 265-274) and figure
- 371 caption.
- L384–396: This section needs thorough revision. The reference to Zhang is
- problematic. Additionally, suggesting a dominant factor may not be valid, as these
- 374 relationships are likely highly location- and case-specific. "Large" is not the right
- word here. Please remind the reader what "#3" refers to.
- 376 **RE:** Thank you so much for the suggestion. We have rewritten this section to remove the
- problematic citation, and avoided implying any universally dominant driver (Lines 437-449).
- We have replaced "large" with "sufficiently frequent" (Line 439). We also clarified that "#3"
- 379 refers to the grain filling stage (Line 447-448). The revised text focused strictly on our own
- 380 stage-specific findings without overgeneralization.
- 381 L417: Use "Different impacts of ... on yields" instead of "yield impacts."
- 382 **RE:** Thank you. We have revised as suggested (Line 470).

- 383 L421: Were these losses shown in any figures or derived from your analysis? If not,
- 384 consider removing this sentence. Also, since the study does not directly assess the
- impact of irrigation, that discussion may not be relevant.
- 386 **RE:** Thank you. The losses have come from our results and we have rewritten there to clear up
- 387 the misunderstanding. (Line 470-471). The descriptions of non-climatic drivers (such as
- irrigation infrastructure) were used to explore possible reasons for the severe yield reductions
- in the Sichuan region and had no relation to our results.
- 390 L437: Replace "rainy stress" with "rain stress."
- 391 **RE:** Thank you. We have revised as suggested (Line 491) and updated this word consistently
- 392 throughout the entire manuscript.
- 393 L456: On what plots are these spatial shifts in concurrent events shown? If you refer
- to shifts over time, clarify this. If not, the sentence is unclear.
- 395 RE: on Figure 3. While figures 1, 2 and 3 indicates the sequence of phenological dates of the
- first, second and third growth stages. We acknowledge the confusion. The term "shifted"
- misleadingly suggests a temporal change; in fact, we intended only to describe the changes in
- 398 the compound heat-drought hotspots by rice growth stage. We have rephrased this passage (in
- both the main text and abstract, Line 321, 400, 510-511) to clearly convey that these are
- 400 spatial distribution characteristics, not temporal shifts.
- L457: "Spatial difference in phenology" is unclear, please rephrase.
- 402 **RE:** We have rephrased this sentence to: "These spatial patterns are driven primarily by
- differences in crop phenology across locations, such as the timing of flowering was earlier in
- the upstream than in the lower Yangtze River basin, rather than by the spatial distribution of
- 405 extreme climate conditions." (Lines 511-513)
- 406 L463 onwards (Conclusion): The conclusion is not the right place to introduce new
- 407 references or discuss limitations. Consider revising this section and relocating these
- 408 points to more appropriate sections in the manuscript.
- 409 **RE:** We have revised the Conclusion section to summarize key findings that have been
- presented and removed any newly cited literature or discussions of limitations. All limitations
- 411 have been moved (Lines 519-524).
- 412 **References:**
- 413 Holly Wang, H., & Zhang, H. (2003). On the Possibility of a Private Crop Insurance Market: A

- Spatial Statistics Approach. Journal of Risk and Insurance, 70(1), 111–124.
- 415 https://doi.org/10.1111/1539-6975.00051
- Li, Z., Liu, W., Ye, T., Chen, S., & Shan, H. (2022). Observed and CMIP6 simulated occurrence
- and intensity of compound agroclimatic extremes over maize harvested areas in China.
- Weather and Climate Extremes, 38, 100503. https://doi.org/10.1016/j.wace.2022.100503
- Luo, Y., Zhang, Z., Chen, Y., Li, Z., & Tao, F. (2020). ChinaCropPhen1km: A high-resolution
- 420 crop phenological dataset for three staple crops in China during 2000–2015 based on leaf
- area index (LAI) products. Earth System Science Data, 12(1), 197-214.
- 422 https://doi.org/10.5194/essd-12-197-2020
- Tootoonchi, F., Sadegh, M., Haerter, J. O., Räty, O., Grabs, T., & Teutschbein, C. (2022).
- 424 Copulas for hydroclimatic analysis: A practice-oriented overview. WIREs Water, 9(2),
- 425 e1579. https://doi.org/10.1002/wat2.1579
- Wu, H., Su, X., & Singh, V. P. (2021). Blended Dry and Hot Events Index for Monitoring Dry-
- Hot Events Over Global Land Areas. Geophysical Research Letters, 48(24),
- 428 e2021GL096181. https://doi.org/10.1029/2021GL096181
- 429 Ye, T., Nie, J., Wang, J., Shi, P., & Wang, Z. (2015). Performance of detrending models of crop
- 430 yield risk assessment: Evaluation on real and hypothetical yield data. Stochastic
- 431 Environmental Research and Risk Assessment, 29(1), 109–117.
- 432 https://doi.org/10.1007/s00477-014-0871-x

---

## Author Response (AR2)

- 1 Title: Spatiotemporal variation of growth-stage specific concurrent climate
- 2 extremes and their yield impacts for rice in southern China
- 3 Response to Report #1:
- 4 Submitted on 03 Sep 2025
- 5 Referee #2: Faranak Tootoonchi, faranak.tootoonchi@gmail.com
- 6 Thanks. The paper has improved. I have a few more comments. Please note that the
- 7 line numbers are related to the version in track-change mode and NOT the clean
- 8 version.
- 9 **RE:** Thank you so much for your comments and suggestions on our manuscript. We have
- responded to the comments and suggestions point-by-point below (in blue). **The line numbers**
- 11 are related to the track-change version.
- 12 **Major Comments:**
- 13 L14: To capture
- **RE:** Thank you. We have revised the sentence accordingly as requested. The new sentence is:
- 15 "Existing studies predominantly rely on the whole growing-season scale and relative thresholds,
- and limit the ability to capture crop physiological sensitivities and yield responses that vary
- critically across growth stages." (Line 15).
- 18 **L204: Stressor**
- 19 **RE:** Thank you. We have revised it as requested (Line 170).
- 20 **L204: This concept**
- 21 **RE:** Thank you. We have revised it as requested (Line 170).
- 22 **L216: The trigger of is unclear.**
- 23 **RE:** Thank you. We have revised the text by replacing "The trigger of" with "The threshold of"
- for clarity (Line 182).
- 25 L216-218: Not sure why we need this sentence about other studies having other
- 26 threshold/cut-off values.
- 27 **RE:** Thank you. We have removed the sentence and retained only the references (Lines 183-
- 28 184).
- 29 L251-261: Consider adding the suitable reference you used to extract these
- 30 steps. https://wires.onlinelibrary.wiley.com/doi/10.1002/wat2.1579 could work.
- 31 **RE:** Thank you so much. We missed this reference (Tootoonchi et al., 2022) in the previous

- 32 version and have now added it (Line 206).
- 33 L264:269: These sentences are not needed in my view. The text is a bit
- encyclopedic and distracts the flow. Consider removing the section but adding
- 35 references in the previous paragraph.
- 36 RE: Thank you for the suggestion. We have removed the corresponding sentences and
- 37 incorporated the relevant references in Line 205. This revision ensures that the section remains
- 38 focused exclusively on methodological content without unnecessary encyclopedic details.
- 39 L281-283: This sentence is unclear to me.
- 40 **RE:** In our definition, if no extreme event occurs in a given year, the severity score is set to "0".
- When fitting copula, we did not include these zero-severity years. However, when calculating
- 42 the joint exceedance probability, these zero values must be considered again to ensure that all
- 43 years (including those without extreme events) are accounted for. We have revised the text into
- a more straightforward description (Lines 226-228).
- Eq 10 and eq 13: Why do you need eq10? Why not to directly use eq13?
- **RE:** The two equations serve different purposes. Equation 10 was used to estimate the trend in
- 47 the historical yield time series, while Equation 13 was to derive the statistical relationship
- between yield anomaly and compound severities. In the previous version, we have mistakenly
- 49 used identical symbols for regression coefficients in the two equations, which was misleading.
- This has been corrected in the revision (Lines 293-296).
- 51 L551:552: Please rewrite the sentence.
- 52 **RE:** Thank you. We have corrected this sentence (Line 439).
- 53 A general comment: Please re-read the paper thoroughly to see if further
- technical language/ sentence modifications are required.
- **RE:** Thank you. We have thoroughly re-read the paper and carefully checked all technical
- language and sentence structures. Corrections and improvements have been made for the lines
- 57 22, 52, 78-79, 104, 106, 141, 144, 359, 401, 404, 412, 469, 480, 504-505, 521.
- 58 **References:**
- Tootoonchi, F., Sadegh, M., Haerter, J. O., Räty, O., Grabs, T., & Teutschbein, C. (2022).
- 60 Copulas for hydroclimatic analysis: A practice-oriented overview. WIREs Water, 9(2),
- e1579. https://doi.org/10.1002/wat2.1579

---

## Author Response (AR3)

- 1 Title: Spatiotemporal variation of growth-stage specific concurrent climate
- 2 extremes and their yield impacts for rice in southern China
- 3 **15 Sep 2025**
- 4 Editor decision: Publish subject to minor revisions (review by editor)
- 5 by Gabriele Messori
- 6 I am pleased to accept your manuscript for publication in Earth System Dynamics
- 7 subject to a final edit. Please update the links that you provide in the data availability
- 8 section to ensure that they do not point to a home page but rather to the actual data
- 9 download page. If this exists in English, then please link to the English version and not
- to the Chinese-language one.
- 11 **RE:** Thank you. We have updated the links provided in the Data Availability section to ensure
- that they now point directly to the actual data download pages (Lines 530-536 in the clean
- version). Where available, we have provided the English versions of the links. For dataset that
- is not accessible through a direct download page, we have removed the corresponding
- statements from the availability section.

---

## Author Response (AR4)

- 1 Title: Spatiotemporal variation of growth-stage specific concurrent climate
- 2 extremes and their yield impacts for rice in southern China
- 3 **15 Sep 2025**
- 4 Public justification (visible to the public if the article is accepted and published):
- 5 by Gabriele Messori
- 6 Thank you for this update. One of the links that you provided is however not working
- 7 correctly: https://doi.org/10.57760/sciencedb.06963
- 8 Please verify this and update the data availability section accordingly before providing
- 9 the final paper production files, after which I am pleased to accept your submission for
- 10 publication in Earth System Dynamics.
- 11 **RE:** Thank you for pointing this out. We have updated the link in the Data Availability
- section. The correct link is:
- https://www.scidb.cn/detail?dataSetId=b07f90ea5f0c4e359fa4119a0030f9da, which is now
- working properly.
- 15 In addition, we have also updated the author affiliations to reflect the recent changes in their
- 16 institutional names.